# Fine-grained Late-interaction Multi-modal Retrieval for Retrieval Augmented Visual Question Answering

**Weizhe Lin, Jinghong Chen,* Jingbiao Mei, Alexandru Coca, Bill Byrne**
Department of Engineering
University of Cambridge
Cambridge, United Kingdom CB2 1PZ
{wl356, jc2124, jm2245, ac2123, wjb31}@cam.ac.uk

## Abstract

Knowledge-based Visual Question Answering (KB-VQA) requires VQA systems to utilize knowledge from external knowledge bases to answer visually-grounded questions. Retrieval-Augmented Visual Question Answering (RA-VQA), a strong framework to tackle KB-VQA, first retrieves related documents with Dense Passage Retrieval (DPR) and then uses them to answer questions. This paper proposes Fine-grained Late-interaction Multi-modal Retrieval (FLMR) which significantly improves knowledge retrieval in RA-VQA. FLMR addresses two major limitations in RA-VQA's retriever: (1) the image representations obtained via image-to-text transforms can be incomplete and inaccurate and (2) relevance scores between queries and documents are computed with one-dimensional embeddings, which can be insensitive to finer-grained relevance. FLMR overcomes these limitations by obtaining image representations that complement those from the image-to-text transforms using a vision model aligned with an existing text-based retriever through a simple alignment network. FLMR also encodes images and questions using multi-dimensional embeddings to capture finer-grained relevance between queries and documents. FLMR significantly improves the original RA-VQA retriever's PRRecall@5 by approximately 8%. Finally, we equipped RA-VQA with two state-of-the-art large multi-modal/language models to achieve $\sim 61\%$ VQA score in the OK-VQA dataset.

## 1   Introduction

Knowledge-based Visual Question Answering (KB-VQA) is a challenging problem that lies at the intersection of Computer Vision, Natural Language Processing, and Information Retrieval. The objective of VQA is to read an image and answer a question related to the image content. KB-VQA poses an additional challenge: in order to answer the question correctly, the system needs to draw on relevant information from an external knowledge source, such as a knowledge graph or a database. Therefore, tackling KB-VQA tasks crucially depends on the ability to retrieve relevant information and to ground the answer generation process in the retrieved knowledge.

Retrieval Augmented Visual Question Answering (RA-VQA) is a framework designed to answer difficult KB-VQA questions [Luo et al., 2021, Gao et al., 2022, Lin and Byrne, 2022], with the most recent version from Lin and Byrne [2022] achieving performance close to large models (such as GPT-3 [Brown et al., 2020]) while using much simpler models. RA-VQA first retrieves $K$ documents relevant to the image and the question from an external knowledge base, and then generates the answer using a Large Language Model (LLM) grounded in the retrieved passages.

---

*Equally contributed as the first author

37th Conference on Neural Information Processing Systems (NeurIPS 2023).

We focus on two major limitations in RA-VQA's retriever. (1) Incomplete image understanding: image representations are obtained via image-to-text transforms such as captioning and object detection. While effective, this approach can result in incomplete image understanding, which hinders the retrieval of relevant knowledge. This is a common issue for retrieval-based KB-VQA systems in the literature. (2) Lossy compression of visual scenes and questions to a single embedding: the Dense Passage Retrieval (DPR) [Karpukhin et al., 2020] retriever, widely used in current retrieval-based QA systems, computes relevance scores between queries and documents with their respective, one-dimensional embeddings. However, compressing complex visual scenes and questions into a single embedding can be lossy. This is especially problematic in KB-VQA, where queries and visual elements are more diverse than in other Open-domain QA tasks. DPR could overlook finer-grained relevance, resulting in degraded retrieval performance.

To address these two limitations we propose an enhanced knowledge retrieval approach called Fine-grained Late-interaction Multi-modal Retrieval (FLMR). FLMR incorporates finer-grained, token-level visual and textual features into multi-dimensional embeddings. When computing relevance scores, FLMR considers the interaction between every pair of token embeddings, including cross-modality interaction between texts and images, enabling a finer-grained assessment of relevance. We also introduce large vision models such as ViT [Dosovitskiy et al., 2021] to produce visual tokens that complement text-based image representations for more complete image understanding. To ensure that the interactions between visual and text tokens are well-defined, we align the vision model with the text-based retriever with a simple yet effective alignment training procedure. We also find that FLMR is able to make use of finer-grained regions of interest, leading to better recall rate, whereas DPR's recall rate degrades when these finer-grained features are incorporated. Our FLMR retriever achieves a significant increase of approximately 8% in PRRecall@5 for knowledge retrieval, and a competitive VQA score of 61%, surpassing the state-of-the-art models with the same scale of parameters.

We summarize our contributions as follows:

- We introduce FLMR, the first-of-its-kind to leverage Late Interaction[2] and multi-dimensional representations to capture fine-grained, cross-modality relevance that significantly improve retrieval performance over existing state-of-the-art KB-VQA retrievers;

- We show that introducing image representations from large vision model after a simple yet effective alignment procedure can complement image representations obtained via image-to-text transforms, leading to more complete image understanding, better knowledge retrieval and higher VQA accuracy. This offers improvements to current VQA systems as many systems have only a single mode of image understanding that relies on either image-to-text transforms or vision models;

- We achieve a substantial improvement of approximately 8% in knowledge PRRecall@5 over other state-of-the-art retrievers in the OK-VQA dataset, with an accuracy of 61% that surpasses other systems with similar parameter sizes.

## 2 Related Work

**Visual Question Answering Systems.** Recent work in VQA can be roughly divided into four categories with respect to multi-modal modeling: (1) Visual and textual features can be fused via cross-modality fusion [Yu et al., 2018, Singh et al., 2019, Yu et al., 2019, Jiang et al., 2020, Guo et al., 2021]; (2) Multi-modal models can be trained from scratch to jointly understand vision and language before they are fine-tuned to perform VQA tasks [Tan and Bansal, 2019, Chen et al., 2020, Gan et al., 2020, Li et al., 2020a, Wang et al., 2022, Zhang et al., 2021, Li et al., 2021]; (3) Vision model and language model that have been pre-trained on uni-modal corpora can be aligned to avoid expensive multi-modal pre-training [Guo et al., 2023, Dai et al., 2022, Singh et al., 2022]. (4) Image-to-text transforms such as captioning can be used to transform images into texts to enable the use of text-only reasoning pipelines [Lin and Byrne, 2022, Gui et al., 2021, Lin et al., 2022, Luo et al., 2021, Yang et al., 2022, Gao et al., 2022, Hu et al., 2022a]. Building on these Vision-and-Language modeling techniques, our work shows that image-to-text transforms and aligned vision models can complement each other to provide more complete visual information, leading to improved performance in both knowledge retrieval and VQA.

---

[2]Dual encoder architecture where the queries and documents are first encoded into token-level embeddings and these embeddings are then aggregated to compute final relevance scores

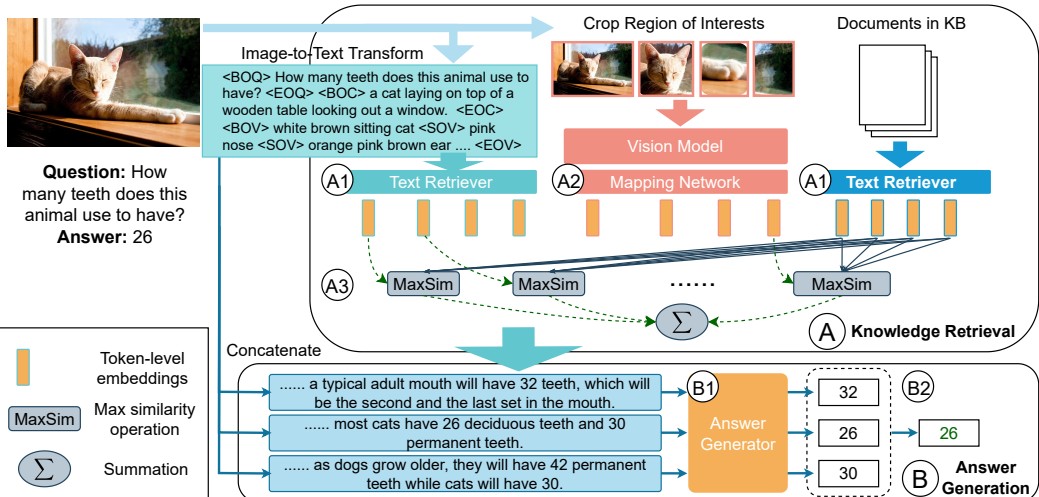

Figure 1: Overview of RA-VQA-v2. The system consists of two steps: (A) Knowledge Retrieval and (B) Answer Generation. (A.1) A text retriever is used to obtain token-level embeddings of text-based vision (obtained by captioning and object detection) and text documents in the database. (A.2) Visual tokens are obtained from the image and the region-of-interest patches using a vision model and a mapping network. (A.3) Relevance score between the query and the document is computed by aggregating the fine-grained relevance at token level with late interaction mechanism (Eq. 12). (B.1) The answer generator takes the text query, the image, and the retrieved documents as input, generating one candidate answer per retrieved document. (B.2) The answer with the highest joint probability is selected.

**Knowledge-based VQA Systems.** Recent KB-VQA systems can access both structured data, such as ConceptNet and other KGs [Narasimhan et al., 2018, Garderes et al., 2020, Li et al., 2020b, Wu et al., 2022, Marino et al., 2021, Chen et al., 2023a], as well as unstructured data such as Wikipedia passages [Wu et al., 2022, Gao et al., 2022, Gui et al., 2021] for knowledge retrieval. LLMs can also be a source of "implicit world knowledge": KAT [Gui et al., 2021] and REVIVE [Lin et al., 2022] prompt GPT-3 to generate potential answer candidates. RA-VQA [Lin and Byrne, 2022] and its prior works [Luo et al., 2021, Qu et al., 2021, Gao et al., 2022] ground answer generation in the retrieved knowledge from external KBs to achieve excellent VQA performance. Our work improves this retriever-reader pipeline with a novel knowledge retriever which significantly improves the recall rate of knowledge retrieval as well as the final VQA performance.

**Knowledge Retrieval.** Most knowledge retrievers in QA systems are based on DPR and its variants [Karpukhin et al., 2020, Gui et al., 2021, Luo et al., 2021, Gui et al., 2021, Lin and Byrne, 2022, Wu and Mooney, 2022]. These mainly use one-dimensional embeddings and contrastive learning for training. Late Interaction models [Khattab and Zaharia, 2020, Santhanam et al., 2022a] have recently achieved state-of-the-art performance on QA knowledge retrieval. Our FLMR extends this paradigm to work with multi-modal features and shows that incorporating finer-grained visual features, such as regions-of-interest, leads to superior retrieval performance. EnFoRe [Wu and Mooney, 2022] retrieves a list of entities from the image, the query, and the answer candidates, and then explicitly learns scores to indicate the importance of each fine-grained entity. FILIP [Yao et al., 2022] has a similar late-interaction setting but it focuses on single modal query (image-text retrieval). To the best of our knowledge, FLMR is also the first to introduce cross-modality, token-level late interactions to compute relevance scores for KB-VQA knowledge retrieval. We also propose a light-weight method that aligns a vision model with a text-based retriever to incorporate more complete multi-modal information in retrieval queries. Compared to previous approaches that rely on expensive pre-training on multi-modal datasets [Chen et al., 2022a, Yao et al., 2022], FLMR's vision-language alignment process is efficient and can be done in 4 hours with one A-100 GPU.

# 3 Method

In this section, we introduce RA-VQA-v2, which builds upon the original RA-VQA framework [Lin and Byrne, 2022] but is equipped with Fine-grained Late-interaction Multi-modal Retriever (FLMR) to enhance knowledge retrieval. As illustrated in Fig. 1, the framework consists of two stages: Knowledge Retrieval (Sec. 3.1) and Answer Generation (Sec. 3.2).

## 3.1 Knowledge Retrieval

The FLMR system consists of two encoders: a vision model $F_V$ and a language model $F_L$ that encode image and textual features, respectively.

**Visual Features.** We utilize two types of visual representations: (1) text-based vision representations (textual description of visual elements) obtained by image-to-text transforms and (2) feature-based vision representations obtained by large vision models.

For text-based vision representations, to allow a direct comparison, we follow Lin and Byrne [2022] to extract objects and their attributes using VinVL [Zhang et al., 2021] and generate image captions using Oscar [Li et al., 2020a]. For each image $I$, we obtain a textual description that contains serialized object names, attributes, and descriptive captions [Lin and Byrne, 2022]. The sequence is appended to the question $q$ to form the query. For simplicity of notation, the question $q$ always includes text-based vision unless otherwise specified.

For feature-based vision representations, we use the vision model $F_V$ to extract both global and regional image feature representations. For regional image feature representations, we further use the object detection results of VinVL to locate $N_{ROI}$ (Region-of-Interest) bounding boxes. To filter bounding box proposals from VinVL, we use the predicted class name associated with each box to select objects explicitly mentioned in the question $q$, and then prioritize bounding boxes with larger areas. Using the vision model $F_V$, we then obtain one global image representation $g = F_V(I) \in \mathcal{R}^{d_V}$ from the image $I$ and ROI-based regional representations $\{r_i = F_V(I_i^p) \in \mathcal{R}^{d_V}\}_{i=1,...,N_{ROI}}$ from the image ROI patches $\{I_i^p : i = 1, ..., N_{ROI}\}$ which contain finer-grained details.

**Token-Level Embeddings.** Compared with DPR's compressed, one-dimensional representation of queries and documents, FLMR preserves richer information by employing token-level, multi-dimensional embeddings to improve retrieval. We obtain token-level embeddings for both textual input and visual input. These are concatenated to form the final embeddings of queries and documents.

To align the vision and text modalities, we train a mapping network $\mathcal{F}_M$ that learns to project visual features from vision model $\mathcal{F}_V$ with hidden size $d_V$ into the latent space of the language model $\mathcal{F}_L$ with hidden size $d_L$. The mapping network, a 2-layer multi-layer perceptron, projects each visual representation into $N_{vt}$ visual tokens, i.e. $\mathcal{R}^{d_V} \to \mathcal{R}^{N_{vt}d_L/2} \to \mathcal{R}^{N_{vt}d_L}$ and finally reshaped into $\mathcal{R}^{N_{vt} \times d_L}$.

Formally, the final query embeddings $\mathbf{Q}$ are:

$$\mathbf{Q} = [\mathcal{F}_L(q), \mathcal{F}_M([g, r_1, r_2, ..., r_{N_{ROI}}])] \in \mathcal{R}^{l_Q \times d_L}, \tag{1}$$

where $l_Q = l_q + (N_{ROI} + 1) \times N_{vt}$. $l_q$ is the length of the question $q$. $[v_1, ..., v_N]$ denotes the concatenation of $N$ embeddings $v_1$ to $v_N$.

The documents in the knowledge base are represented by embeddings $\mathbf{D}$ obtained from the document content $d$ of length $l_D$:

$$\mathbf{D} = \mathcal{F}_L(d) \in \mathcal{R}^{l_D \times d_L}, \tag{2}$$

**Multi-Modal Late Interaction.** We compute the relevance score between a question-image pair $\bar{\mathbf{q}} = (q, I)$ and a document $d$ by a late interaction formula similar to that in ColBERT but under the multi-modal context:

$$r(\bar{\mathbf{q}}, d) = r((q, I), d) = \sum_{i=1}^{l_Q} \max_{j=1}^{l_D} \mathbf{Q}_i \mathbf{D}_j^\top \tag{3}$$

For each query token, the MAX operation selects the highest relevance score over all document tokens. In preliminary experiments, other operations (e.g. MEAN or SUM) were found to be overly sensitive to the length of documents, which can be as short as a single sentence. We note that [PAD]

may dominate in the final score for short documents, whereas longer documents have an inherent advantage due to their greater number of meaningful tokens.

In contrast to DPR, FLMR allows full interactions between every query embedding vector $\mathbf{Q}_i$ and every document embedding vector $\mathbf{D}_j$. FLMR retriever also supports retrieving multi-modal documents. We leave the formulation and results to Appendix H.

**Training and Inference.** To train the model, we treat documents $d^*$ that contain the ground-truth answer to question $q$ as gold (positive) documents. We use in-batch negative sampling for training following Karpukhin et al. [2020]. All documents in a training batch other than $d^*$ are considered negative for $q$, denoted as $\mathcal{N}(q)$. We train with the contrastive loss $\mathcal{L}_{CL}$ over the dataset $\mathcal{D}$:

$$\mathcal{L}_{CL} = - \sum_{(q,d^*)\in\mathcal{D}} \log \frac{\exp\left(r(q,d^*)\right)}{\exp\left(r(q,d^*)\right) + \sum_{z\in\mathcal{N}(q)} \exp\left(r(q,z)\right)} \tag{4}$$

After training, all documents are indexed using PLAID [Santhanam et al., 2022b], which enables fast late-interaction retrieval with a time cost similar to that of DPR.

**Training the Mapping Network for Vision-Language Alignment.** Directly fine-tuning the two models $\mathcal{F}_V$ and $\mathcal{F}_L$ on the retrieval task leads to performance degradation at the start of training, as the models are not yet aligned. Inspired by CLIP [Radford et al., 2021], where a language model is trained to align with a vision model, we align $\mathcal{F}_V$ and $\mathcal{F}_L$ in the context of knowledge retrieval by pre-training the parameters of the mapping network $\mathcal{F}_M$ with a retrieval task.

Given ground-truth image-document pairs $\{(I_p, d_p)\}$, which can be Wikipedia images and their accompanying texts, the system is trained to retrieve the document $d_p$ associated with the input image $I_p$. The relevance between the input image $I$ and a document $d$ is formulated as

$$\mathbf{Q} = \mathcal{F}_M(F_V(I)) \in \mathcal{R}^{N_{vt}\times d_L}; \quad \mathbf{D} = \mathcal{F}_L(d) \in \mathcal{R}^{l_D\times d_L}; \quad r(I,d) = \sum_{i=1}^{N_{vt}} \max_{j=1}^{l_D} \mathbf{Q}_i \mathbf{D}_j^\top \tag{5}$$

where only the parameters of the mapping network $\mathcal{F}_M$ are trained with the contrastive loss in Eq. 4. We provide details of pre-training in Appendix E and discuss its effectiveness in Sec. 5.2.

**Knowledge Retrieval.** We extract top-$K$ documents from the knowledge base as relevant knowledge. The retrieval probability is defined below following the notation of Lin and Byrne [2022] and Lewis et al. [2020]:

$$p_\theta(d_k|\bar{\mathbf{q}}) = \frac{\exp(r(\bar{\mathbf{q}}, d_k))}{\sum_{j=1}^K \exp(r(\bar{\mathbf{q}}, d_j))} \tag{6}$$

where $\theta$ denotes the model parameters of $\mathcal{F}_V, \mathcal{F}_L$, and $\mathcal{F}_M$.

## 3.2 Answer Generation

In principle, the knowledge retrieved by FLMR can be used by any VQA answer generator. We denote the answer generator as $\mathcal{F}_A$ with parameters $\phi$. Following Lin and Byrne [2022], RA-VQA-v2 generates an answer for each retrieved document and selects the best candidate by the joint probability of retrieval and answer generation:

$$\{d_k\}_{k=1}^K = \mathrm{topK}_d\left(p_\theta(d|\bar{\mathbf{q}})\right); \quad \widehat{y}, \widehat{d} = \arg\max_{y,d_k} p(y, d_k|\bar{\mathbf{q}}) = \arg\max_{y,d_k} p_\phi(y|\bar{\mathbf{q}}, d_k)\, p_\theta(d_k|\bar{\mathbf{q}}) \tag{7}$$

The training loss of the answer generator follows that of the underlying model. For example, when using BLIP 2 [Li et al., 2023], we use the cross-entropy loss of the generated sequences:

$$\mathcal{L} = \sum_{(\bar{\mathbf{q}},\mathcal{S})\in\mathcal{T}} \sum_{k=1}^K \log p_\phi(s_k^*|\bar{\mathbf{q}}, d_k) \tag{8}$$

where $\mathcal{T}$ is the whole dataset. $\mathcal{S}$ is the set of human responses. $s_k^* \in \mathcal{S}$ is the answer string that appears in the retrieved document $d_k$, or the most popular answer string[3] if an exact match cannot be found in the document.

---

[3] The most popular answer is the one chosen by most annotators.

# 4 Experiment Setup

**Datasets.** We focus on the OK-VQA dataset where a large portion of questions requires external knowledge (either commonsense or domain-specific) to answer. There are no annotations of 'ground-truth' documents for OK-VQA questions. We follow the literature to use pseudo-relevance labels (a binary indicator of whether a document contains the answer string) as document annotations. We do not evaluate A-OKVQA [Schwenk et al., 2022], a successor of OK-VQA, as it emphasizes visually-grounded reasoning rather than knowledge retrieval. To validate the effectiveness of our proposed approach, we test the retrieval abilities using 2 different corpora, whose statistics can be found in Appendix C:

(1) *Google Search Corpus for OK-VQA* [Luo et al., 2021]: a passage corpus collected for answering OK-VQA questions. Previous work has shown that the corpus is effective for OK-VQA [Luo et al., 2021, Lin and Byrne, 2022]. We use this corpus in evaluating VQA performance since it covers more knowledge for answering the OK-VQA questions.

(2) *Wikipedia Corpus for OK-VQA*: we collect this corpus by gathering all Wikipedia passages on common objects and concepts (e.g. umbrella, dog, hat) and those containing any of the potential answers in OK-VQA training set. This ensures the corpus covers useful knowledge for answering OK-VQA questions. We note that the collected corpus encompasses a multitude of semantically-diverse documents (>100,000) that challenge the retrieval system to identify actually useful documents. For example, all Wikipedia documents with the word 'party' are included in the corpus, ranging from descriptions of fairy tales to political parties. We also use 10% of the WIT dataset [Srinivasan et al., 2021], a corpus based on Wikipedia with image-text pairs, to train the mapping network for multi-modal alignment.

We evaluate on two additional KB-VQA datasets to demonstrate FLMR's generalizability.
(1) FVQA [Wang et al., 2017]: We follow RAVQA [Lin and Byrne, 2022] to preprocess the data. All knowledge triplets are serialized into text sequences to form the knowledge base for retrieval. The average of 5 cross-validation splits is reported.
(2) Infoseek [Chen et al., 2023b]: Infoseek is a newly proposed KB-VQA dataset that provides Wikipedia documents that can be used in answering its questions. We follow Chen et al. [2023b] in preprocessing. First, we remove questions whose answers cannot be found in the provided Wikipedia passages. Second, in additional to the documents covered in the dataset (~60,000), we include less relevant passages to form a knowledge base for retrieval (~100,000 documents). The test set annotation has not been released, and so we split the official validation set again into validation and test sets (~5200 questions).

**Training Setup.** We use `ColBERTv2` [Santhanam et al., 2022a] and CLIP ViT-base [Radford et al., 2021] to initialize the text-based retriever and vision encoder. For the DPR baseline, we use the official DPR checkpoints to initialize the retriever. In answer generation, we use T5-large [Raffel et al., 2020] and BLIP2-Flan-T5-XL. We use 1 Nvidia A100 (80G) for all experiments. We give detailed training hyperparameters in Appendix E. We use LoRA [Hu et al., 2022b] to fine-tune RA-VQA-v2 (BLIP 2) on 1 single GPU. The vision model is frozen throughout all experiments. During vision-language alignment training, only the mapping network is trainable. In training the answer generator, the retriever is frozen. Our implementations are released at https://github.com/LinWeizheDragon/Retrieval-Augmented-Visual-Question-Answering.

**Evaluation.** We present the metrics used to assess the generated answer and the performance of our knowledge retriever below. All reported numbers are averaged from 3 runs with different seeds. We verified the significance of all mentioned improvements with `scipy.stats.ttest_ind` ($p < 0.05$).

(1) *VQA Score*: To evaluate VQA performance, we use the official VQA Score [Marino et al., 2019] which assigns score to the generated answer based on its exact occurrence count in the set of human responses $\mathcal{S}$:

$$\text{VQAScore}(y, \mathcal{S}) = \min\left(\frac{\#_{\mathcal{S}}(y)}{3}, 1\right), \tag{9}$$

where $\#_{\mathcal{S}}(y)$ is the occurrence of $y$ in human responses $S$. This score ensures that a model is partially rewarded even if it generates a less popular answer among the human responses [Luo et al., 2021].

(2) *Exact Match (EM)* awards point if any of the annotated answers is generated exactly: $\text{EM}(y, \mathcal{S}) = \min(\#_{\mathcal{S}}(y), 1)$ .

(3) *Pseudo Relevance Recall (PRRecall@K)*: To evaluate the retriever, we adopt pseudo relevance following Luo et al. [2021] due to the absence of ground-truth knowledge documents for each query. A document is considered pseudo-relevant if it contains any human-annotated answers. PRRecall@K measures whether the retrieved $K$ documents contain at least one pseudo-relevant document: PRRecall@K $= \min \left( \sum_{k=1}^{K} H(d_k, \mathcal{S}), 1 \right)$, where $H(d_k, \mathcal{S})$ evaluates to 1 if the retrieved document $d_k$ contains any answer in $\mathcal{S}$, and 0 otherwise. The metric is averaged across the test set.

**Baselines.** To demonstrate the effectiveness of **FLMR**, we take a **DPR** retriever as a baseline. In later sections, FLMR *w/o Late Interaction* refers to the corresponding DPR baseline. We use the same training data and hyper-parameters to build a multi-modal retriever based on DPR. For fair comparison, we keep the product $N_{vt} \times d_L$ identical for DPR and FLMR so that they have the same number of parameters in the mapping networks. Since DPR can only handle one-dimensional query and document embeddings, we sum the embeddings of the [CLS] token from $\mathcal{F}_L(\cdot)$ and the visual tokens from $\mathcal{F}_M(\mathcal{F}_V(\cdot))$ to reduce the dimension to $1 \times d_L$ (details in Appendix D).

We also compare our VQA performance with the latest KB-VQA models. Amongst these models, ConceptBERT [Garderes et al., 2020], KRISP [Marino et al., 2021], VRR [Luo et al., 2021], MAVEx [Wu et al., 2022], KAT-T5 [Gui et al., 2021], TRiG-Ensemble [Gao et al., 2022], and RA-VQA are relatively small in model size (<1B), whereas PICa [Yang et al., 2022], KAT [Gui et al., 2021], Prophet [Shao et al., 2023], PromptCap [Hu et al., 2022a], REVIVE [Lin et al., 2022], PALI [Chen et al., 2022b], Flamingo [Alayrac et al., 2022], PaLM-E [Driess et al., 2023] use very large models such as GPT-3 (175B) and PaLM-E (562B).

# 5  Results and Key Findings

## 5.1  VQA Performance

As shown in Table 1, recent models leveraged LLM or Large Multi-modal Models (LMMs) to achieve excellent performance on OK-VQA. The best performance to date is by PaLM-E, achieving a VQA score of 66.1 with 562 billion pre-training parameters. The original RA-VQA formalism achieves a lower VQA Score of 54.48 but with only 800 million parameters.

We first compare RA-VQA-v2 (with FLMR retrieval) with RA-VQA (with DPR retrieval). Our replication of RA-VQA (T5-Large) (Table 1, Row 22) achieves similar PRRecall@5 and VQA Score as the published results of RA-VQA (Table 1, Row 8). Compared with RA-VQA (T5-large), RA-VQA-v2 (T5-large) improves the PRRecall@5 significantly from 83.08% to 89.32%, leading to a gain of 3.4 in VQA Score (51.45 to 54.85). This suggests that improvement in knowledge retrieval benefits answer generation via retrieval augmentation.

We also show the effectiveness of knowledge augmentation by comparing the underlying base models with their retrieval-augmented version. As shown, T5-large and BLIP 2 (fine-tuned with OK-VQA data) achieve 47.52 and 55.44 VQA Scores, respectively. Their retrieval-augmented version, RA-VQA-v2 (T5-large) and RA-VQA-v2 (BLIP 2) gain 7.33 and 6.64 in VQA Score, respectively. For readers' interest, we provide more thorough analyses on the performance that the underlying answer generator model attains and the gain brought by knowledge retrieval in Appendix I.

To confirm that text-based vision can aid LMMs such as BLIP 2 which already has its own image encoder in VQA tasks, we remove text-based vision from RA-VQA-v2 (BLIP 2) and BLIP 2 (fine-tuned). This results in a decrease in VQA performance from 62.03 to 60.37 and 55.44 to 54.10, respectively (Table 3), suggesting that text-based vision contains useful information not included in the visual features obtained by BLIP 2's own image encoders.

RA-VQA-v2 achieves comparable and even better performance when compared with systems that use very large ($\geq$13B parameters) LLMs and LMMs. With BLIP 2 ($\approx$3B), RA-VQA-v2 outperforms Flamingo (80B) by 4.19 VQA Score. It also outperforms many recent systems that use GPT-3 (175B) as an answer generator or knowledge source, such as PromptCap, REVIVE, and KAT. It achieves similar performance to that of PALI (17B) (62.03 vs 64.5 VQA Score). With comparable parameter sizes, RA-VQA-v2 (BLIP 2, 3B) outperforms PALI (3B) by a large absolute margin (62.03 vs 52.40 VQA Score). We emphasize that RA-VQA-v2 can be used in conjunction with virtually any existing LLMs and LMMs to offer substantial improvements, as demonstrated by the T5-large and BLIP 2 experiments.

Table 1: Model Performance on OK-VQA. Knowledge Source abbreviations: C: ConceptNet; W: Wikipidia; GS: GoogleSearch; GI: Google Images. EM stands for Exact Match. VQA stands for VQA Score. R stands for PRRecall. The best performance in literature is underlined.

| # | Model | Base Models | K | Knowl. Src. | R@5 | EM | VQA |
|---|---|---|---|---|---|---|---|
| 1 | ConceptBERT | | | C | | | 33.66 |
| 2 | KRISP | | | C + W | | | 38.35 |
| 3 | VRR | | 100 | GS | | | 45.08 |
| 4 | MAVEx | | | W + C + GI | | | 39.40 |
| 5 | KAT-T5 | T5-large | 40 | W | | | 44.25 |
| 6 | TRiG-Ensemble | T5-large | 100 | W | | 54.73 | 50.50 |
| 7 | RA-VQA (joint training) | T5-large | 50 | GS | 82.84 | 59.41 | 54.48 |
| 8 | RA-VQA | T5-large | 5 | GS | 81.25 | 55.77 | 51.22 |
| | *Systems based on large models ($\geq$3B parameters)* | | | | | | |
| 9 | PICa | GPT-3 | | GPT-3 | | | 48.00 |
| 10 | KAT-Ensemble | T5-large, GPT-3 | 40 | W + GPT-3 | | | 54.41 |
| 11 | Prophet | GPT-3 | | GPT-3 | | | 61.11 |
| 12 | PromptCap | GPT-3 | | GPT-3 | | | 60.40 |
| 13 | REVIVE | GPT-3 | | W + GPT-3 | | | 58.00 |
| 14 | PALI | PALI (3B) | | PALI | | | 52.40 |
| 15 | PALI | PALI (15B) | | PALI | | | 56.50 |
| 16 | PALI | PALI (17B) | | PALI | | | 64.50 |
| 17 | Flamingo | Flamingo (80B) | | Flamingo | | | 57.80 |
| 18 | PaLM-E | PaLM-E (562B) | | PaLM-E | | | 66.10 |
| | *Baselines without knowledge retrieval* | | | | | | |
| 19 | T5-large (fine-tuned) *w/o knowledge* | T5-large | | | | 51.38 | 47.52 |
| 20 | BLIP 2 (fine-tuned) *w/o knowledge* | BLIP 2 (T5-XL) | | | | 59.49 | 55.44 |
| | *Our proposed models (models w/o Late-interaction use DPR instead of FLMR)* | | | | | | |
| 21 | RA-VQA-v2 (T5-large) | T5-large | 5 | GS | 89.32 | 58.85 | 54.85 |
| 22 | *w/o ROI & VE & Late-interaction* | T5-large | 5 | GS | 83.08 | 55.89 | 51.45 |
| 23 | RA-VQA-v2 (BLIP 2) | BLIP 2 (T5-XL) | 5 | GS | **89.32** | **62.01** | **62.08** |
| 24 | *w/o ROI* | BLIP 2 (T5-XL) | 5 | GS | 87.02 | 61.63 | 60.75 |
| 25 | *w/o ROI & VE* | BLIP 2 (T5-XL) | 5 | GS | 85.99 | 59.95 | 60.41 |
| 26 | *w/o Late-interaction* | BLIP 2 (T5-XL) | 5 | GS | 82.90 | 59.00 | 58.20 |
| 27 | *w/o ROI & Late-interaction* | BLIP 2 (T5-XL) | 5 | GS | 83.43 | 60.18 | 59.21 |
| 28 | *w/o ROI & VE & Late-interaction* | BLIP 2 (T5-XL) | 5 | GS | 83.08 | 59.49 | 58.70 |

## 5.2 Retrieval Performance

**Text-based/Feature-based Vision.** As shown in Table 2, previous retrievers (RA-VQA, VRR) achieve ≈82.84% PRRecall@5 using only text-based vision (textual descriptions of visual scenes). We show that visual features obtained via aligned vision models (feature-based vision) are equally effective as text-based vision. Relying on questions only, FLMR has a baseline retrieval score of 74.81 PRRecall@5. Incorporating text-based vision and feature-based vision increase PRRecall@5 to 85.99 and 85.08, respectively. Furthermore, feature-based vision provides information complementary to test-based vision, as demonstrated by the better PRRecall@5 at 87.02 when the two are combined. The same trend is observed for DPR-based retrieval system, though less dramatically (from 83.08 to 83.43). We note that pre-training the mapping network for vision-language alignment is crucial for good performance. Without such pre-training, performance degrades to 85.71. These results confirm that incorporating aligned vision encoders in the retrieval process compensates for the information loss in image-to-text transforms.

**Effects of Late Interaction and ROIs.** Late Interaction enables FLMR to capture fine-grained relevance of token-level embeddings. As shown in Table 2, using the same query and document representations, upgrading DPR to FLMR leads to consistent improvement in retrieval performance by a large margin up to ~6% (comparing Table 2 Row 8 & 13).

Table 2: Retrieval performance on Google Search (GS) and Wikipedia. Text-based vision refers to textual descriptions of images (such as OCR, caption, objects and attributes). Feature-based vision is obtained using a neural vision model directly (e.g. ViT). R@K refers to PRRecall@K.

| # | Retriever | Text-based Vision | Feature-based Vision | GS R@5 | GS R@10 | Wikipedia R@5 | Wikipedia R@10 |
|---|---|---|---|---|---|---|---|
| 1 | VRR | ✓ | - | 80.4 | 88.55 | | |
| 2 | RA-VQA-FrDPR | ✓ | - | 81.25 | 88.51 | | |
| 3 | RA-VQA | ✓ | - | 82.84 | 89.00 | | |
| 4 | DPR | - | - | 73.11 | 82.05 | 57.03 | 69.84 |
| 5 | DPR | ✓ | - | 83.08 | 89.77 | 66.04 | 75.94 |
| 6 | DPR | - | ✓ | 80.52 | 88.27 | 65.84 | 75.85 |
| 7 | DPR | ✓ | ✓ | 83.43 | 90.31 | 66.88 | 76.35 |
| 8 | DPR | ✓ | ✓+9ROIs | 82.90 | 89.95 | 65.86 | 75.90 |
| 9 | FLMR | - | - | 74.81 | 83.10 | 57.20 | 70.11 |
| 10 | FLMR | ✓ | - | 85.99 | 92.79 | 66.50 | 76.80 |
| 11 | FLMR | - | ✓ | 85.08 | 91.80 | 66.90 | 77.05 |
| 12 | FLMR | ✓ | ✓ | 87.02 | 92.69 | 67.50 | 77.60 |
| 13 | FLMR | ✓ | ✓+9ROIs | **89.32** | **94.02** | **68.10** | **78.01** |
| 14 | *w/o alignment pre-training* | ✓ | ✓+9ROIs | 85.71 | 92.41 | 66.40 | 76.10 |

Table 3: Removing text-based vision from answer generation reduces the VQA performance, showing that text-based vision offers more complete image understanding.

| Model | VQA Score |
|---|---|
| RA-VQA-v2 (BLIP 2) | 62.03 |
| *w/o text-based vision* | 60.37 |
| BLIP 2 (fine-tuned) *w/o knowledge* | 55.44 |
| *w/o text-based vision* | 54.10 |

Table 4: Comparison of ROI selection methods. R stands for PRRecall.

| | R@5 | R@10 |
|---|---|---|
| 4 Object-centric ROIs | 88.01 | 93.62 |
| 4 Random ROIs | 86.9 | 93.20 |
| 4 Evenly-split ROIs | 86.96 | 93.16 |

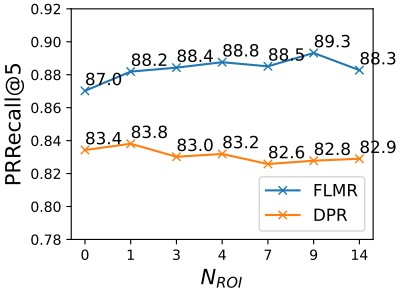

Figure 2: PRRecall@5 versus the number of ROIs. Finer-grained ROIs cause performance degradation in DPR, while FLMR captures them to improve retrieval performance.

In addition to token-level relevance, FLMR can utilize fine-grained Region-of-Interest (ROI) features with Late Interaction whereas DPR can not. This can be demonstrated by Fig. 2: as the number of ROIs increases, DPR performance degrades. This may be because DPR's one-dimensional query and document embeddings are not expressive enough to encompass fine-grained details of the ROI visual cues. As shown in Table 2 and Table 1 Row 27-28, adding more ROIs effectively adds noise which adversely impacts the retrieval performance (83.43 to 82.9), and in turn worsen VQA scores (59.2 to 58.2).

**Object-centric ROIs improve retrieval.** We also conduct ablation studies to show that the performance improvements brought by ROIs come from the finer-grained information captured by them rather than from increases in the number of features. We compare FLMR with 4 object-centric ROIs (obtained by object detection) against 2 baseline ROI selection methods: (1) randomly crop 4 patches of size larger than $100 \times 100$ from the image as ROIs; (2) evenly split the image to obtain the top-left, top-right, bottom-left, and bottom-right of the image as ROIs. As shown in Table 4, FLMR with 4 ROIs from VinVL object detection outperforms others, suggesting that it is the object-centric, fine-grained ROIs that improve the performance.

**Retrieval performance on FVQA and Infoseek.** As shown in Table 5, we observed similar improvements with FLMR. FLMR with both text- and feature-based vision improves DPR by 2.3%

Table 5: Retrieval performance on FVQA [Wang et al., 2017] and Infoseek [Chen et al., 2023b]. Average recall on 5 splits is reported for FVQA. FLMR outperforms DPR trained with the same data with a clear margin.

|  | FVQA Recall@5(Std.) | Infoseek Recall@5 |
| --- | --- | --- |
| DPR | 68.58(0.01) | 44.88 |
| FLMR (Visual Encoder) | 70.88(0.01) | 46.42 |
| FLMR (Visual Encoder & 10 ROIs) | **72.37**(0.01) | **47.08** |

Figure 3: Selected query tokens connected by document tokens that have the highest token-level relevance with them, as computed by FLMR. For example, amongst all document tokens, '26' and '30' have the highest relevance with the query token 'how' and 'many', respectively. This shows that FLMR can capture fine-grained document relevance. Zoom in for better visualization.

and 1.54% Recall@5 on FVQA and Infoseek, respectively. Incorporating ROI features further improves its performance to 72.37 on FVQA and 47.08 on Infoseek. This suggests that FLMR is generalizable to other KB-VQA retrieval tasks and can bring steady improvements relative to baselines.

**Qualitative analysis of FLMR retrieval**. Figure 3 shows FLMR retrieval in action. The orange lines connect the query token and the document token with the highest relevance score, which will be preserved after the MaxSim operation and will contribute to the final retrieval score. We can see that token-level interaction indeed captures fine-grained relevance between the query and the document. For example, the retriever recognizes that the numbers "26" and "30" in the document are highly relevant to "How many" in the query. We can also see that the image tokens are aligned with the text tokens: the image tokens corresponding to the cat (IMG, ROI3 and ROI4) point to the words "cats" and "cat" in the document. This demonstrates the effectiveness of vision-language alignment that gives rise to explainable cross-modality relevance. We provide more case studies in Appendix J.

# 6 Conclusion

In this paper, we proposed Fine-grained Late-interaction Multi-modal Retrieval (FLMR), the first of its kind to leverage fine-grained token-level relevance between queries and documents for VQA tasks. FLMR incorporates feature-based vision using an aligned vision model that complements text-based vision to enhance image understanding, improve retrieval performance and advance VQA performance. We achieve superior performance in OK-VQA, greatly surpassing previous systems with similar parameter size and closes the gap with those systems utilizing very large (>13B) models.

# 7 Acknowledgement

Weizhe Lin was supported by a Research Studentship funded by Toyota Motor Europe (RG92562(24020)). We thank our colleagues, Daniel Olmeda Reino (Toyota Motor Europe) and Jonas Ambeck (Toyota Motor Europe), who provided insight and expertise in this project.

Prof. Bill Byrne holds concurrent appointments as a Professor of Information Engineering at Cambridge University and as an Amazon Scholar. This publication describes work performed at Cambridge University and is not associated with Amazon.

We would also like to thank all the reviewers for their knowledgeable reviews.

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

## A   Limitations

We chose the Google Search corpus [Luo et al., 2021] for our question-answering system as it provides good coverage of the knowledge needed and is publicly available. However, as noted by the authors of RA-VQA, additional knowledge bases may be required to answer some questions correctly. Future work may address the issue by improving the quality and expanding the coverage of knowledge.

## B   Ethics Statement

We do not perceive any immediate ethical concerns associated with the misuse of our proposed system. There is a possibility that the trained KB-VQA system might generate inappropriate or biased content as a result of the training data biases during LLM and LMM pre-training and fine-tuning. Therefore, it is advised to conduct an ethical review prior to deploying the system in live service.

## C   Data Statistics

Table 6 shows the data statistics of the OK-VQA dataset. Table 7 displays the number of passages in the document collections used for evaluating the retrieval systems. Note that the WIT corpus is introduced in Appendix H, which is used for investigating the retrieval of multi-modal documents.

Table 6: OK-VQA dataset statistics.

| Category | Number |
| --- | --- |
| train questions | 9,009 |
| valid questions | 5,046 |
| images | 14,055 |

Table 7: Data statistics of document collections used in retrieval.

| Corpus | # of passages |
| --- | --- |
| GS for OK-VQA [Luo et al., 2021] | 168,306 |
| Wikipedia for OK-VQA | 114,637 |
| WIT for OK-VQA (Appendix H) | 87,419 |

## D   Details of DPR baselines

We build a **DPR** retriever as a baseline for FLMR. We apply the same pre-training strategy, training data, and hyperparameters to construct a multi-modal retriever based on DPR. Particularly, we keep the product $N_{vt} \times d_L$ and the number of parameters of the vision mapping networks identical for FLMR and DPR for a fair comparison. Since DPR can only handle one-dimensional query and document embeddings, we sum the embeddings of the [CLS] token from $\mathcal{F}_L(\cdot)$ and the visual tokens from $F_M(F_V(\cdot))$ to reduce the dimension to $1 \times d_L$. Formally, the query and document embeddings are:

$$\mathbf{Q_{dpr}} = \left( \mathcal{F}_{L,\text{CLS}}(q) + \mathcal{F}_M(\mathcal{F}_V(g)) + \sum_{i=1,\dots,N_{ROI}} \mathcal{F}_M(\mathcal{F}_V(r_i)) \right) \in \mathcal{R}^{d_L},$$

$$\mathbf{D_{dpr}} = \mathcal{F}_{L,CLS}(d) + \mathcal{F}_M(\mathcal{F}_V(I_d)) \in \mathcal{R}^{d_L}. \tag{10}$$

where $I_d$ is the image of the document if multi-modal document collection is used and otherwise omitted. The inner product search (supported by FAISS [Johnson et al., 2019]) is used to train and retrieve documents with DPR.

## E   Training and Hyperparameter Details

We use `ColBERTv2` and `openai/clip-vit-base-patch32` to initialize the text-based retriever and vision encoder. For the DPR baseline, we use `facebook/dpr-single-nq-base` to initialize the retriever. In answer generation, we use `t5-large` and `Salesforce/blip2-flan-t5-xl`.

With `openai/clip-vit-base-patch32`, $d_V = 768$. For FLMR, we use $N_{vt} = 32$ visual tokens per image representation and $d_L = 128$. For DPR, we use $N_{vt} = 6$ and $d_L = 768$ so that the number of parameters of vision mapping network is similar to that of FLMR: $N_{vt} \times d_L \sim 128 \times 32$. The mapping network consists of two fully-connected layers with tanh activation. The output of last layer is reshaped into $N_{vt} \times d_L$ visual tokens. Other model parameters are: $l_q = 512$, $l_d = 512$. $N_{ROI} = 9$ unless otherwise specified.

We use 1 Nvidia A100 (80G) for all experiments. The optimizer is Adam [Kingma and Ba, 2015]. In training the retrievers, we use learning rate $10^{-4}$, batch size 30, gradient accumulation steps 2 for 10k steps (for both DPR and FLMR retrievers). When training RA-VQA-v2 (T5-large), we use learning rate $6 \times 10^{-5}$, batch size 2, gradient accumulation 16 for up to 20 epochs. We use a linearly-decaying scheduler to reduce learning rate from $6 \times 10^{-5}$ to 0 after 20 epochs. We use LoRA [Hu et al., 2022b] to train RA-VQA-v2 (BLIP2) with learning rate $10^{-4}$, batch size 4, gradient accumulation steps 16 for up to 6k steps. LoRA is configured to use the default huggingface-PEFT setting: `r=8`, `lora_alpha=32, lora_dropout=0.1`.

The vision model is frozen throughout all experiments. In pre-training the mapping network, only the mapping network is trainable. When training the answer generator, the retriever is frozen.

We report the required GPU hours on 1 Nvidia A100 (80G): for vision-language alignment of retrieval models, approximately 4 GPU hours are needed. Training the FLMR retriever requires around 12 GPU hours (10k steps) including the time of running testing after training is complete. Training RA-VQA-v2 (BLIP 2) with LoRA requires around 12 GPU hours (6k steps) including the time of running validation per 1k steps. Training the RA-VQA-v2 (T5-large) requires around 12 GPU hours (3k steps) including the time of running validation every 500 steps.

All implementations are released at https://github.com/LinWeizheDragon/Retrieval-Augmented-Visual-Question-Answering.

## F   Artifacts and License

We list the resources used and their License below:

(1) huggingface-transformers (Apache License 2.0) provides pre-trained model checkpoints for BLIP 2, DPR, T5, and their tokenizers: https://github.com/huggingface/transformers

(2) PLAID engine and ColBERTv2 checkpoints (MIT License): https://github.com/stanford-futuredata/ColBERT

(3) FAISS [Johnson et al., 2019] (MIT License) is used to index document embeddings for fast retrieval with DPR: https://github.com/facebookresearch/faiss

(4) huggingface-PEFT (Apache License 2.0) for parameter-efficient LoRA fine-tuning: https://github.com/huggingface/peft

(5) The official RA-VQA implementation (GNU General Public License v3.0): https://github.com/LinWeizheDragon/Retrieval-Augmented-Visual-Question-Answering.

## G   Computational Cost

We report the computational cost in this section.

Table 8: Training and indexing time for FLMR and DPR. Training batch size is 30. The corpus for counting the indexing time is the Google Search Corpus for OK-VQA.

|  | train per 1000 steps | indexing time |
| --- | --- | --- |
| FLMR | 1.2h | 0.28h |
| *w/o ROI* | 1h | 0.25h |
| *w/o ROI & VE* | 0.7h | 0.24h |
| DPR | 0.5h | 0.2h |

Though Late Interaction allows rich interactions at token level and outperforms DPR by a large margin, it also introduces additional latency in retrieval. As shown by Table 8, the training time of FLMR is increased from 0.5h to 0.7h when late interaction is introduced. This latency increase comes from the more complicated token-level loss. When Vision Encoder (VE) and ROI (Region of Interest) are added, the time cost is increased to 1h and 1.2h respectively due to the additional trainable parameters of the mapping network. However, the indexing time does not increase significantly when VE and ROI are added to the FLMR retriever. We note that the FLMR spends slightly more time to build the search index when compared to DPR because an extra clustering step by PLAID [Santhanam et al., 2022b] is required to conduct fast retrieval.

Table 9: Training and inference time of the whole system. Please note that passages are dynamically retrieved, and thus the training and inference time already takes the retrieval latency into account. Batch size is set to 1 for both training and inference time. *w/o ROI & VE* means removing the vision encoder in FLMR.

| Retriever | Generator | Training Speed (iterations/sec) | Inference Speed (iterations/sec) |
|---|---|---|---|
| FLMR | T5-large | 1.16 | 1.11 |
| DPR | T5-large | 1.73 | 1.67 |
| FLMR | BLIP 2 | 1.24 | 0.98 |
| FLMR (w/o ROI & VE) | BLIP 2 | 1.43 | 1.00 |
| DPR | BLIP 2 | 2.14 | 1.30 |

When FLMR is integrated into the full VQA pipeline (we take the BLIP 2 version for example), it reduces the training speed from 2.14 iterations/sec to 1.24 iterations/sec (42%) since the retrieval process is run on the fly. However, in retrieval, the inference speed is only reduced from 1.3 iterations/sec to $\sim$1.0 iterations/sec, which is still affordable when considering the performance boost. The major computational cost remains that of training the answer generator with a great number of parameters.

## H    Retrieving Multi-modal Documents with FLMR

We additionally show that our proposed FLMR system can also be used to retrieve multi-modal documents. Since this is not the focus of our paper, we present the investigation in this appendix.

**Dataset.** We select a subset from WIT [Srinivasan et al., 2021], a knowledge corpus based on Wikipedia where the images associated with the documents are also present, to make an image-text corpus for retrieval. We adopt the same selection process as for the Wikipedia corpus introduced in Sec. 4. The dataset statistics is shown in Table 7.

**Multi-Modal Late Interaction.** We upgrade the document embedding process to accommodate the document image. The documents in the knowledge base are represented by embeddings $\mathbf{D}$ which are obtained from the document content $d$ and its associated image $I_d$:

$$\mathbf{D} = [\mathcal{F}_L(d), \mathcal{F}_M(\mathcal{F}_V(I_d))] \in \mathcal{R}^{l_D \times d_L}, \tag{11}$$

where $l_D = l_d + N_{vt}$, and $l_d$ is the length of the document $d$.

We compute the relevance score between a question-image pair $\bar{\mathbf{q}} = (q, I)$ and a document $\bar{\mathbf{d}} = (d, I_d)$ as follows:

$$r(\bar{\mathbf{q}}, \bar{\mathbf{d}}) = r((q, I), (d, I_d)) = \sum_{i=1}^{l_Q} \max_{j=1}^{l_D} \mathbf{Q}_i \mathbf{D}_j^\top \tag{12}$$

**Discussion.** Both query and document embeddings are multi-modal. Since the same image/text encoder is used to encode images $I, I_d$ and texts $q, d$, respectively. Image-wise and text-wise relevance contribute to the final relevance score; After cross-modality alignment, the vision encoder $\mathcal{F}_M(\mathcal{F}_V(\cdot))$ should produce image embeddings close to the text embeddings produced by $F_L(\cdot)$ in the latent space if the image is relevant to the question, thereby taking the relevance between $I, d$ and $q, I_d$ into account during knowledge retrieval.

As shown in Table 10, the retrieval scores see a slight improvement when document images are also considered (from text-only to multi-modal). This suggests that FLMR supports retrieving multi-modal documents.

However, we note that the gain of incorporating images is marginal. This is because WIT is a strongly text-driven knowledge base as the images are already captioned by human experts. The surrounding texts of images are already dense and informative, which can be searched by FLMR easily. By manual inspection, we also notice that it is very rare that OK-VQA questions seek a document that can only be found by its accompanying images. This also explains the marginal gain we have observed.

In conclusion, we show that FLMR can also be applied to retrieve multi-modal documents, although more challenging questions and better datasets are needed to fully exploit its potential. We leave this as future work.

Table 10: FLMR performance when retrieving documents in WIT. Models suffixed by 'uni-modal' only encode document texts, while 'multi-modal' variants encode document images with vision encoders.

| Model | PRRecall@5 | PRRecall@10 |
|---|---|---|
| DPR-text-only | 68.24 | 77.13 |
| DPR-image-only | 46.29 | 57.70 |
| DPR-multi-modal | 68.78 | 77.90 |
| FLMR-text-only | 72.63 | 81.52 |
| FLMR-image-only | 45.75 | 57.92 |
| FLMR-multi-modal | **73.65** | **81.89** |

# I  Effects of Retrieved Knowledge

Table 11: Comparing Hit Success Rate of RA-VQA-v2 and RA-VQA.

| | Hit Success Rate |
|---|---|
| RA-VQA-v2 (BLIP2) | 9.38 |
| RA-VQA (BLIP2) | 7.86 |
| RA-VQA-v2 (T5-large) | 17.62 |
| RA-VQA (T5-large) | 15.01 |

It is important to understand the task performance that a base model has attained and the gains from knowledge retrieval. We use the official evaluation metrics from RA-VQA: the **Hit Success Ratio (HSR)** which counts questions that cannot be answered by the base VQA model alone and thus require external knowledge to answer.

$$HSR = \mathbb{1}\big\{\widehat{y} \in \mathcal{S} \land \widehat{y}_{NK} \notin \mathcal{S}\big\}; \tag{13}$$

where $y_{NK}$ denotes the generated answer from a fine-tuned base model when no relevant knowledge is provided. HSR reflects the net value of incorporating external documents into answer generation. We can conclude from Table 11 that RA-VQA-v2 steadily improves the HSR of RA-VQA by $\sim 2\%$, showing that the gains in VQA performance come from improved knowledge retrieval. We also observe that T5-large, as an earlier language model, relies more heavily on retrieved knowledge ($>15$ HSR). This is because the base language model of BLIP 2, Flan-T5-XL, is significantly stronger and is able to answer more questions without the aid of external knowledge. This suggests that KB-VQA performance can be improved by either (1) applying stronger base VQA answer generation models, and (2) collecting knowledge documents of higher quality.

We also conduct experiments while increasing $K$ in Table 1 and find that the system performance improves gradually until a saturation point. We notice that the saturation point of FLMR is at around $K = 10$ while that of DPR is at $K = 20$. This suggests that the useful documents are clustered around higher ranks in FLMR compared to DPR.

Table 12: Performance improvements with increasing number of retrieved documents.

|  | K | 5 | 10 | 20 | 50 |
|---|---|---|---|---|---|
| DPR + T5-large | VQA Score | 51.5 | 51.8 | 52.3 | 52.1 |
|  | Recall | 83.08 | 89.77 | 94.05 | 97.25 |
| FLMR + T5-large | VQA Score | 54.9 | 55.3 | 55.4 | 55.4 |
|  | Recall | 89.32 | 94.02 | 96.87 | 98.67 |

## J    Case Study

A case study is presented in Fig. 4. It compares the model outputs and provides expalanations to each case.

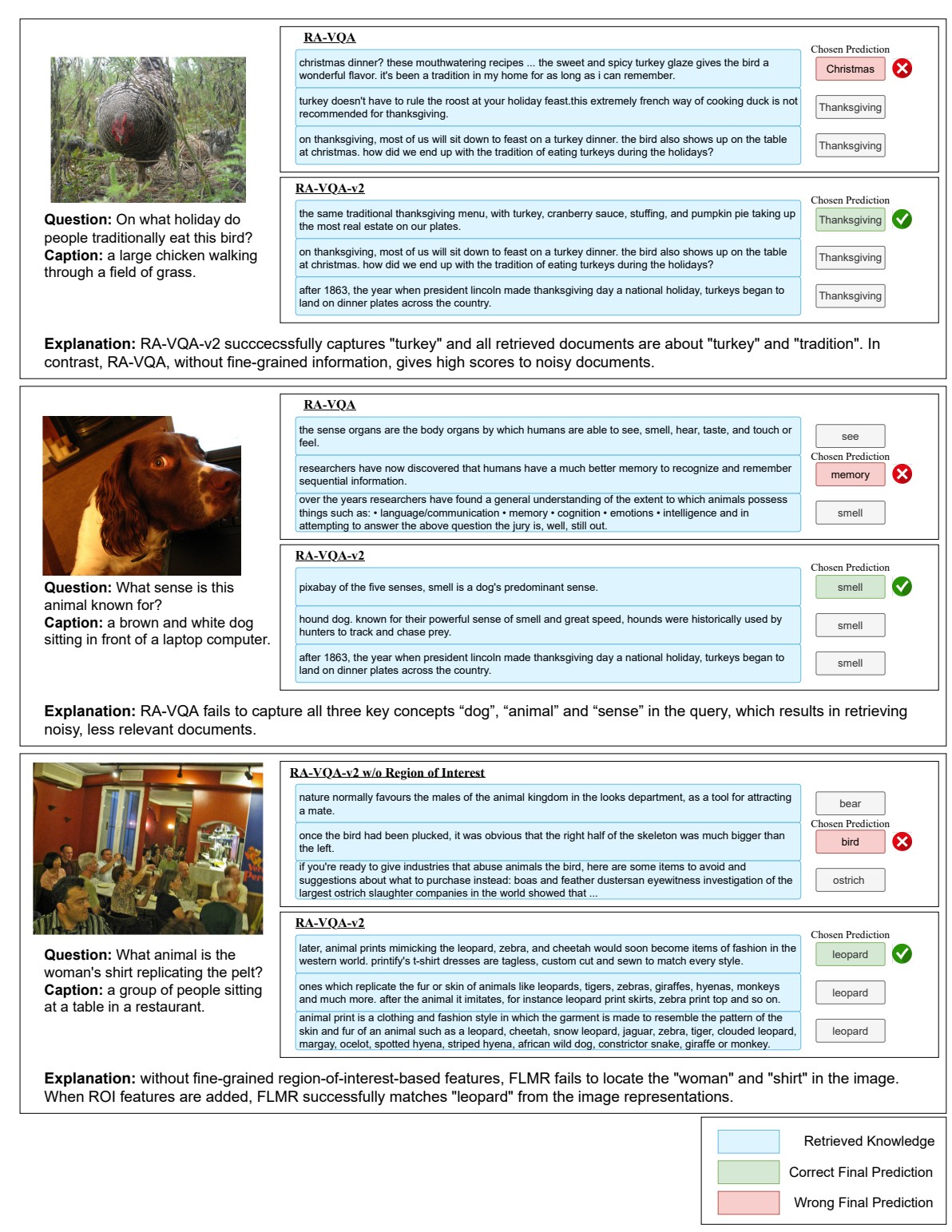

Figure 4: Case study comparing some model variants. Explanations are given to each case. Please zoom in for the best visualization.

