**RA-VQA**

christmas dinner? these mouthwatering recipes ... the sweet and spicy turkey glaze gives the bird a wonderful flavor. it's been a tradition in my home for as long as i can remember.

turkey doesn't have to rule the roost at your holiday feast.this extremely french way of cooking duck is not recommended for thanksgiving.

on thanksgiving, most of us will sit down to feast on a turkey dinner. the bird also shows up on the table at christmas. how did we end up with the tradition of eating turkeys during the holidays?

Chosen Prediction
Christmas ✗
Thanksgiving
Thanksgiving

**RA-VQA-v2**

the same traditional thanksgiving menu, with turkey, cranberry sauce, stuffing, and pumpkin pie taking up the most real estate on our plates.

on thanksgiving, most of us will sit down to feast on a turkey dinner. the bird also shows up on the table at christmas. how did we end up with the tradition of eating turkeys during the holidays?

after 1863, the year when president lincoln made thanksgiving day a national holiday, turkeys began to land on dinner plates across the country.

Chosen Prediction
Thanksgiving ✓
Thanksgiving
Thanksgiving

**Question:** On what holiday do people traditionally eat this bird?
**Caption:** a large chicken walking through a field of grass.

**Explanation:** RA-VQA-v2 succcecssfully captures "turkey" and all retrieved documents are about "turkey" and "tradition". In contrast, RA-VQA, without fine-grained information, gives high scores to noisy documents.

**RA-VQA**

the sense organs are the body organs by which humans are able to see, smell, hear, taste, and touch or feel.

researchers have now discovered that humans have a much better memory to recognize and remember sequential information.

over the years researchers have found a general understanding of the extent to which animals possess things such as: • language/communication • memory • cognition • emotions • intelligence and in attempting to answer the above question the jury is, well, still out.

see
Chosen Prediction
memory ✗
smell

**RA-VQA-v2**

pixabay of the five senses, smell is a dog's predominant sense.

hound dog. known for their powerful sense of smell and great speed, hounds were historically used by hunters to track and chase prey.

after 1863, the year when president lincoln made thanksgiving day a national holiday, turkeys began to land on dinner plates across the country.

Chosen Prediction
smell ✓
smell
smell

**Question:** What sense is this animal known for?
**Caption:** a brown and white dog sitting in front of a laptop computer.

**Explanation:** RA-VQA fails to capture all three key concepts "dog", "animal" and "sense" in the query, which results in retrieving noisy, less relevant documents.

**RA-VQA-v2 w/o Region of Interest**

nature normally favours the males of the animal kingdom in the looks department, as a tool for attracting a mate.

once the bird had been plucked, it was obvious that the right half of the skeleton was much bigger than the left.

if you're ready to give industries that abuse animals the bird, here are some items to avoid and suggestions about what to purchase instead: boas and feather dustersan eyewitness investigation of the largest ostrich slaughter companies in the world showed that ...

bear
Chosen Prediction
bird ✗
ostrich

**RA-VQA-v2**

later, animal prints mimicking the leopard, zebra, and cheetah would soon become items of fashion in the western world. printify's t-shirt dresses are tagless, custom cut and sewn to match every style.

ones which replicate the fur or skin of animals like leopards, tigers, zebras, giraffes, hyenas, monkeys and much more. after the animal it imitates, for instance leopard print skirts, zebra print top and so on.

animal print is a clothing and fashion style in which the garment is made to resemble the pattern of the skin and fur of an animal such as a leopard, cheetah, snow leopard, jaguar, zebra, tiger, clouded leopard, margay, ocelot, spotted hyena, striped hyena, african wild dog, constrictor snake, giraffe or monkey.

Chosen Prediction
leopard ✓
leopard
leopard

**Question:** What animal is the woman's shirt replicating the pelt?
**Caption:** a group of people sitting at a table in a restaurant.

**Explanation:** without fine-grained region-of-interest-based features, FLMR fails to locate the "woman" and "shirt" in the image. When ROI features are added, FLMR successfully matches "leopard" from the image representations.

Retrieved Knowledge
Correct Final Prediction
Wrong Final Prediction

Figure 1: Case study comparing some model variants. Explanations are given to each case. Please zoom in for the best visualization.