# OpenReview forum: "Fine-grained Late-interaction Multi-modal Retrieval for Retrieval Augmented Visual Question Answering"
_NeurIPS.cc/2023/Conference — NeurIPS 2023 poster_

### Official Review · Reviewer_jgsy · 2023-06-14

**Soundness:** 2 fair
**Presentation:** 3 good
**Contribution:** 2 fair
**Rating:** 4
**Confidence:** 4

**Summary:**

This paper studies two problems in knowledge-based VQA: 1) the image input is often transformed into the text; 2) the relevance score between query and document is estimated based on one-dimensional dot product.
To address these problems, the authors leverage several image features, including ROIs features from ViT and text descriptions.
The experiments are conducted on the OK-VQA benchmark dataset.
After being equipped with the recent strong BLIP v2 model, the proposed method achieves very promising results.

**Strengths:**

- The paper is well-written and easy to follow for most parts.
- The two studied problems are very important and practical for knowledge-based VQA models.
- The proposed method achieves improved model performance after combining with BLIP v2.

**Weaknesses:**

- Is there also ground-truth knowledge for the questions in the test set?
- With the first question, did the authors use ground-truth knowledge for testing?
- In fact, one dataset is not fairly convincing and the A-OKVQA should also be selected as a benchmark dataset. The reason that the authors provided does not make much sense.
- The ablation study in Table 1 is very confusing. For example, there are no detailed experiments for T5-Large. Combining line 26 and 27, it seems that ROI&VE poses a negative effect on model performance.
- For Eqn.3 and Eqn.5, why use max rather than mean or sum?
- The ```late interaction``` in this paper means?
- Line 141 confuses me a lot. It seems that the authors first map each ROI image to one feature, and then transform it to $N_{vt}$ tokens. Why don't we just use the $N_{vt}$ token features from ViT?

**Questions:**

The proposed method achieves very promising results on the OK-VQA benchmark.
However, several concerns refrain me from holding a positive view of this paper.
I would like to see the authors' rebuttal to these concerns.

---

> ### Author Rebuttal · Authors · 2023-08-09
>
> Thank you very much for raising these concerns.
>
> - **Is there also ground-truth knowledge for the questions in the test set? Did the authors use ground-truth knowledge for testing?**
>     - No and no. There is no ground-truth knowledge annotation of the document collection for the question in the test set. To assess retrieval performance, we adhere to established norms in the literature, using pseudo-relevance (the presence of the answer within the retrieved document) as our metric of evaluation [1]. We also assess the ultimate VQA performance, which acts as a measure of the effectiveness of the retrieved documents in aiding the QA process.
> - **In fact, one dataset is not fairly convincing and the A-OKVQA should also be selected as a benchmark dataset.**
>     - Thank you for raising this concern. Please see the general response for reports on additional datasets which we will include in the final version.
>     - As a research focusing on knowledge retrieval in VQA, we note that A-OKVQA is not suitable for our purpose as only a small number of questions (~18%) in the dataset require outside knowledge. This was reported by the dataset authors [2], and we have confirmed it in our own studies. Therefore, a knowledge retrieval system cannot be properly evaluated on this dataset.
> - **There are no detailed experiments for T5-Large.**
>     - We validate the efficiency of FLMR paired with BLIPv2, currently recognized as the SOTA, allowing for fair comparisons with the most recent literature. We report the result using T5-large since it was used in the original RA-VQA. In our experiments, we found that the trends seen with BLIPv2 also hold for T5-large. We were constrained by space and, consequently, these results were not incorporated. We share the results here:
>
>         | Model | R@5 | EM | VQA |
>         | --- | --- | --- | --- |
>         | RA-VQA-v2 (T5-large) | 89.32 | 58.85 | 54.85 |
>         | w/o ROI | 87.02 | 57.80 | 53.75 |
>         | w/o ROI & V | 85.99 | 57.65 | 53.78 |
>         | w/o Late Interaction | 82.90 | 55.30 | 50.95 |
>         | w/o ROI & Late-interaction | 83.43 | 56.20 | 52.03 |
>         | w/o ROI & VE & Late-interaction | 83.08 | 55.89 | 51.45 |
>
> - **it seems that ROI&VE poses a negative effect on model performance.**
>     - Thanks for your careful reading. When using the DPR approach, introducing VE alone leads to a slight improvement. However, the performance declines with the further introduction of ROIs. Row 5 and Row 7 in Table 2 show that adding VE improves retrieval performance. However, the performance is reduced when ROI features are incorporated (comparing Row 7 and 8). This regression in performance can be attributed to DPR's less effective one-dimensional representation of all the ROIs compared to FLMR's late interaction. We will add one more line in Table 1 to demonstrate the point more clearly: DPR with VE and without ROI: Recall@5=83.43, EM=60.18, VQA Score=59.21. The result, compared to Table 1 line 27, further confirms that when using DPR, adding VE is beneficial but adding ROIs adversely impacts performance. This observation in fact motivates our approach, as we believe finer-grained information should benefit retrieval. In our work, we show that FLMR’s late-interaction mechanism allows for better use of the ROI features, leading to improved retrieval and VQA performance (Table 1, Row 23&24).
> - **For Eqn.3 and Eqn.5, why use max rather than mean or sum?**
>     - The MAX operation selects the highest relevance score between a query token and all doc tokens. In our preliminary experiments, MEAN/SUM exhibits instability when dealing with a wide range of document lengths, from extremely short pieces (a sentence) to very long ones. We note that [PAD] tokens in short documents may dominate in the final score, while longer documents have an inherent advantage due to their greater number of meaningful tokens. Using SUM/MEAN in practice reduces the performance by ~3% Recall@5. Therefore, we use MAX as a straightforward and optimal solution. We will add this explanation in the final version.
> - **The `late interaction` in this paper means?**
>     - In this paper, it refers to the dual encoder architecture where the queries and documents are first encoded into token-level embeddings by Transformers. These embeddings are then aggregated to compute final relevance scores. This setup not only utilizes dense encoders but also allows full interaction. You can find a very comprehensive demonstration in Figure 2 of the ColBERT paper [3]. We will make this clear in revision.
> - **Line 141 confuses me a lot. It seems that the authors first map each ROI image to one feature, and then transform it to $N_{vt}$ tokens. Why don't we just use the $N_{vt}$ token features from ViT?**
>     - Only the first [CLS] token of ViT has been trained in its pre-training process. As a result, only ViT’s first token embeddings are meaningful. We tried in preliminary experiments to use $N_{vt}$ tokens from ViT. The model takes an extremely long time (50k steps) to converge, and the final performance is much lower (-8 Recall@5) than using the first token alone. We found that extracting regional features around objects for use with ViT’s first token embeddings resulted in readily trainable regional representations. In addition, the original ViT cannot handle object-centric ROIs since it splits an input image into evenly-split regions. We also show that object-centric ROIs are more beneficial than evenly-split regions (Line 298-305).
>
>
> We hope that our clarifications have addressed your concerns, and we kindly invite you to consider the value and contributions of our work in your final assessment.
>
> [1] Retrieval Augmented Visual Question Answering with Outside Knowledge.
>
> [2] A-OKVQA: A Benchmark for Visual Question Answering using World Knowledge
>
> [3] ColBERT: Efficient and Effective Passage Search via Contextualized Late Interaction over BERT.

---

> > ### Comment · Reviewer_jgsy · 2023-08-22
> >
> > Thank the authors for their explanations of the usage of knowledge during testing and supplemented experiments.
> > But I still believe that A-OKVQA still needs to be experimented on.
> > I decide to raise my original score.

---

### Official Review · Reviewer_shyj · 2023-07-07

**Soundness:** 4 excellent
**Presentation:** 3 good
**Contribution:** 3 good
**Rating:** 6
**Confidence:** 4

**Summary:**

The paper proposed a fine-grained Late-interaction multimodal knowledge retriever for knowledge-based VQA. The core idea to the to both visual tokens and question tokens as token-level queries to retrieve document in late interaction fashion. The paper suggests that (1) squeezing query representation and document representation into one vectors hinder the retrieval performance. (2) fine-grained visual representation helps retrieval. The paper also show that the framework can be easily extended to retrieve multimodal documents

**Strengths:**

(1) The paper  provide a recipe to take advantage of late Interaction and multi-dimensional representations to capture fine-grained, cross-modality relevance to improve retrieval performance that further boost the (A)OK-VQA tasks

(2) The paper perform very thorough analysis on the retrieval results using different corpus choices for OKVQA tasks (Google search and wikipedia)

(3) The paper is well-written and easy to follow

(4) The paper provide 2 baseline VQA model (T5 and BLIP2) as the answer generator and the results looks very promising

(5) The paper also report the additional inference time cost in the appendix indicating that the inference time over head is not un acceptable

**Weaknesses:**

Overall,  I think it is a solid paper, here is some minor comments

(1) While the inference time cost has been mentioned, it is also good to mention storage cost as you need to storage the features for each tokens. Hope to know if it requires very good SSD or RAM though it is cheaper than the GPUs

(2) in the bottom block in table 1, I hope to understand the performance difference when the number of knowledge sentences grows. It is good to plot the performance curve with x axis denoting the number of sentences and the y shows the vqa score. This can help us understand (a) performance-knowledge saturation speed compared to DPR, showing we need to use the FLMR if we have enough bucket to hold more knowledge as inputs.


(3) [1] also uses fine-grained visual information as multiple queries to retrieve knowledge and it is good to compare to that as well







------------------------------------
post rebuttal: I read the author's rebuttal and still willing to support the paper as my initial rating



References:
[1] Entity-Focused Dense Passage Retrieval for Outside-Knowledge Visual Question Answering

**Questions:**

Please comment on the weakness part

**Limitations:**

The  authors adequately addressed the limitations

---

> ### Author Rebuttal · Authors · 2023-08-09
>
> Thank you very much for your very positive review. Thank you for your careful reading as well. We respond to your constructive suggestions as below:
>
> 1. **Hope to know if it requires very good SSD or RAM though it is cheaper than the GPUs.**
>
>     Thank you for raising this point. Our cluster employs commonly-seen SSDs and RAMs. The index file which indexes the Google Search Corpus (~170k documents) is 23 GB. A more complete analysis of the tradeoff between speed and storage can be found in [2]. We will add as much detail as possible to the camera-ready version for readers’ interest.
>
> 2. **I hope to understand the performance difference when the number of knowledge sentences grows.**
>
>     Performance-knowledge saturation is indeed interesting for understanding the behavior of FLMR and DPR. Following your suggestion, we conduct experiments increasing K in Table 1 and find that the system performance first improves gradually until a saturation point. The results are as follows:
>
>
>     |  | K | 5 | 10 | 20 | 50 |
>     | --- | --- | --- | --- | --- | --- |
>     | DPR + T5-large | VQA Score | 51.5 | 51.8 | 52.3 | 52.1 |
>     |  | Recall | 83.08 | 89.77 | 94.05 | 97.25 |
>     | FLMR + T5-large | VQA Score | 54.9 | 55.3 | 55.4 | 55.4 |
>     |  | Recall | 89.32 | 94.02 | 96.87 | 98.67 |
>
>     We notice that the saturation point of FLMR is at around K=10 while that of DPR is at K=20. This suggests that the useful documents are clustered around higher ranks in FLMR compared to DPR.
>
>     We will provide such a figure in the main content or in the appendices if accepted. Thanks for your help in improving our paper.
>
> 3. **EnFoRe [1] also uses fine-grained visual information as multiple queries to retrieve knowledge and it is good to compare to that as well**
>
>     Thank you for your suggestions. We will compare our system to EnFoRe in the camera-ready version.
>
>     Our work and EnFoRe share a common motivation: entities’ information can be crucial in retrieving knowledge relevant to the question. However, our approaches are quite different. The two main differences are:
>     - First, EnFoRe retrieves a list of entities from the image, the query, and the answer candidates. EnFoRe then explicitly learns scores to indicate the importance of each entity. In comparison, FLMR mainly utilizes an object detection model to extract object-centric ROIs and entity information. We employ late-interaction to learn the interaction between regional image features and document token embeddings implicitly. After training, our system also learns interpretable retrieval as depicted in Figure 3.
>     - Second, in terms of the retriever formulation, they employ one-dimensional embeddings following DPR. The scores from text embeddings and entity embeddings are added, similar to the baseline system DPR (VE+9ROIs) in Row 8, Table 2. In contrast, we employ token-level multi-dimensional embeddings for both image and text, which have been demonstrated to be more effective by comparing Row 8 and Row 13 of Table 2.
>
>     We can also compare performance directly. Using Wikipedia passages, EnFoRe achieves ~34.88% Precision@5. In comparison, our system achieves ~46.41% Precision@5 when using Wikipedia passages. Please note that we use the scores from their paper directly instead of replicating their system.
>
>
> Finally, thank you very much for your careful reading, and we are happy that you have read the appendices thoroughly as well. Thank you!
>
> [1] Entity-Focused Dense Passage Retrieval for Outside-Knowledge Visual Question Answering
>
> [2] PLAID: An Efficient Engine for Late Interaction Retrieval

---

### Official Review · Reviewer_xLzi · 2023-07-08

**Soundness:** 4 excellent
**Presentation:** 4 excellent
**Contribution:** 4 excellent
**Rating:** 7
**Confidence:** 3

**Summary:**

This paper proposes a multi-vector (late interaction) multi-modal retrieval-based approach for Knowledge-based Visual Question Answering, inspired by the ColBERT multi-vector representations for text. The retriever improves the recall substantially over prior multi-modal methods that use single-vector encoding. The approach obtains token-level embeddings for both textual input and visual input. They also align the vision and text modalities, by training a mapping network. The paper achieves very strong SoTA scores on OK-VQA.

**Strengths:**

1. The paper's experiments are intuitive and extensive. They experiment test two types of visual representations (text-based and feature-based), aligning vision and text modalities, multi-modal late interaction, etc. in a series of very instructive ablations.

2. The results are very strong and convincing, aligned with the strong intuitions offered in the writing.

3. The paper offers nice, though minimalistic, qualitative analysis of their (nicely interpretable) retrieval.

**Weaknesses:**

1. What are the connections to FILIP (Yao et al., 2021), which is also a late interaction retriever for some multi-modal settings. FILIP seems to tackle zero-shot image classification and image-text retrieval, which are much simpler tasks perhaps, but the connection must be explored in the paper.

2. The authors do report quite a few ablations for the retriever and the full method. That's great. But it's not necessarily great to report so many results on the test set! It might be better in the future to report such ablations on a validation set.

**Questions:**

n/a

---

> ### Author Rebuttal · Authors · 2023-08-09
>
> Thank you for your very careful reading and the accurate summarization of our strengths. Thank you very much!
>
> 1. **What are the connections to FILIP (Yao et al., 2021), which is also a late interaction retriever for some multi-modal settings. FILIP seems to tackle zero-shot image classification and image-text retrieval, which are much simpler tasks perhaps, but the connection must be explored in the paper.**
>
>     Thank you for your suggestion. We will add discussions about this work in the camera-ready version. We provide a detailed discussion for your interest.
>
>     FILIP and FLMR are similar in two aspects:
>
>     - First, both systems are dual-stream models: FILIP and FLMR encode text and image modalities with separate encoders and learn aligned, joint embedding spaces for the two modalities.
>     - Second, both systems explicitly model the interactions between token-level embeddings to capture fine-grained similarity/relevance. In a broad sense, both systems make use of late interaction in aggregating token-level interactions, keeping the computation tractable, though the mathematical formulations are not exactly the same.
>
>     The difference between FILIP and FLMR are three folds.
>
>     - First, as you mentioned, the task of FILIP is finding related texts with images. Their query is single modal (either image or text). The text being matched is usually concise and straightforward. In contrast, in our task, the query consists of not only images but also questions and text-based vision (image captioning, OCR, detected objects), leading to long and complex multi-modal queries. This is one of our motivations to employ token-level embeddings to allow full interactions between complex multi-modal queries and long knowledge documents.
>     - Second, FILIP pre-trained the whole system from scratch for text-image alignment. Conversely, our strategy is predicated upon aligning any current vision encoder (including ViT, CLIP ViT) with any late interaction text retriever (for example, ColBERT, ColBERTv2 and so on), making the retrieval performance grow with the development of general-purpose vision encoders and late-interaction text retrievers. As an additional benefit, we find that only 4 GPU hours are needed for alignment.
>     - Third, the formulations for fine-grained image features are different. We use object-centric regional image features, while FILIP uses evenly-split regional features. We demonstrate in Line 298-305 that object-centric ROIs achieved better performance than evenly-split regional features. We believe that FILIP can also benefit from our formulation.
>
>      We will cite FILIP and discuss the connections in the Related Work section in the final version if accepted. Thank you for your knowledgeable help in improving our paper.
>
> 2. **The authors do report quite a few ablations for the retriever and the full method. That's great. But it's not necessarily great to report so many results on the test set! It might be better in the future to report such ablations on a validation set.**
>
>     Thank you very much for your suggestion. In our future work, we will follow your suggestion to report ablation more rigorously.
>
> Finally, thank you for your valuable support and thorough analysis for our work! Thank you for your careful review.

---

> > ### Comment · Reviewer_xLzi · 2023-08-17
> >
> > Thank you for the response. My rating is a 7 (accept) and I will keep that.

---

### Official Review · Reviewer_XziC · 2023-07-23

**Soundness:** 3 good
**Presentation:** 3 good
**Contribution:** 2 fair
**Rating:** 5
**Confidence:** 4

**Summary:**

This paper introduces a new method, Fine-grained Late-interaction Multi-modal Retrieval, to improve knowledge retrieval in RA-VQA. The method addresses two major limitations in RA-VQA’s retriever:  incomplete image understanding and lossy compression of visual scenes and questions.

**Strengths:**

1) The paper proposes to capture fine-grained, cross-modality relevance that significantly improves retrieval performance.
2) The paper introduces a simple yet effective alignment procedure that can complement image representations obtained via image-to-text transforms, leading to more complete image understanding, better knowledge retrieval, and higher VQA accuracy.
3) The method improves approximately 8% in knowledge PRRecall@5 over other state-of-the-art retrievers in the OK-VQA dataset.


**Weaknesses:**

1) Method. This work used the VinVL approach, which leverages regions of interest for visual features. I could not pinpoint the main contributions of this work, which DPR does not make. For a better understanding of the differences between the two methods, perhaps the authors could present them side-by-side in a figure.

2) FLMR and DPR. Why FLMR preserves richer information better than DPR representations? Because it also uses regions of the image with object class labels for predictions? I understand that FLMR allows full interaction between every query embedding vector and every document embedding vector. Is this the only difference?

3) In-batch negative sampling. Has DPR also used in-batch negative sampling? If not, did you try to compare when DPR uses in-batch negative sampling? How significant is this to the proposed method?

4) Results. Regarding Table 1, I am surprised to see that the comparison against BLIPv2 is not that large, while the proposed method uses the BLIPv2 caption generator. If PALI would use BLIPv2 text decoder, would it be better? I am unsure whether the main contribution is related to the BLIPv2/T5 text decoder or to better modeling.


Overall, I appreciate the paper as it presents good results, but the contributions are largely unclear. I am open to the authors' feedback and other reviewers' opinions.



**Questions:**

I wrote above.

**Limitations:**

I wrote above.

---

> ### Author Rebuttal · Authors · 2023-08-09
>
> Thank you for your careful reading and kind suggestions. We will answer your questions point by point below:
>
> 1. **This work used the VinVL approach, which leverages regions of interest for visual features. I could not pinpoint the main contributions of this work, which DPR does not make. For a better understanding of the differences between the two methods, perhaps the authors could present them side-by-side in a figure.**
>
>     Our work uses VinVL to identify regions of interest, from which regional visual features are subsequently extracted using ViT. VinVL can be replaced by any other better object detectors in principle. Our contribution lies in the enhancement of the DPR retriever through the application of multi-dimensional embeddings and late interaction processes within the context of multi-modal retrieval, which is tangential to VinVL.
>
>     To allow a fair comparison, we add dense image features and ROI features to both FLMR and DPR in our ablation study (Table 2, Row 8 and Row 13). The DPR retriever with VE and ROI features follows the formulation in Appendix D. We find that the performance gain of FLMR is mainly derived from the multi-dimenstional late interaction mechanism, rather than the addition of global and regional visual features.
>
> 2. **FLMR and DPR. Why FLMR preserves richer information better than DPR representations? Because it also uses regions of the image with object class labels for predictions? I understand that FLMR allows full interaction between every query embedding vector and every document embedding vector. Is this the only difference?**
>
>     Preserving token-level embeddings and allowing full interaction is indeed key to FLMR’s richer information. We rigorously investigate how FLMR compares with DPR as follows:
>
>     First, token-level embeddings with late-interaction already improve the retrieval performance without dense visual features, as both the document and the query (which consists of not only questions, but also a series of detected objects, image captions, and OCR results) can be long in length, calling for fine-grained relevance computation. We demonstrate this by comparing FLMR and DPR in a setting where dense image features are not provided to either system (DPR: Row 5, Table 1; FLMR: Row 10, Table 1).
>
>     Second, FLMR’s multi-dimensional representation and late interaction mechanism makes it possible to utilize region-of-interest features obtained from a vision encoder. However, we observe that DPR failed to utilize fine-grained ROI features (Line 288-297). This is because DPR’s one-dimensional query embeddings are not expressive enough to encompass fine-grained details of the ROI visual cues. As the number of ROIs increases, DPR’s performance is negatively impacted (Figure 2). In contrast, FLMR allows each ROI regional feature to interact with each of document embeddings, achieving increasingly better performance as the number of ROI increases (Figure 2). Other reviewers (xLzi) have found Figure 3 helpful in demonstrating that each ROI feature in FLMR is meaningfully connected to document token embeddings.
>
> 3. **In-batch negative sampling. Has DPR also used in-batch negative sampling? If not, did you try to compare when DPR uses in-batch negative sampling? How significant is this to the proposed method?**
>
>     Yes. DPR used in-batch negative sampling. For a fair comparison, we strictly applied the same training configuration to FLMR and DPR (Line 155-156, Appendix D). In practice, in-batch negative sampling improves 1~2% Recall@5 for both systems.
>
> 4. **Results. Regarding Table 1, I am surprised to see that the comparison against BLIPv2 is not that large, while the proposed method uses the BLIPv2 caption generator. If PALI would use BLIPv2 text decoder, would it be better? I am unsure whether the main contribution is related to the BLIPv2/T5 text decoder or to better modeling.**
>
>     In brief, we did not use BLIPv2 as the caption generator, although BLIPv2 and T5 are used as the answer generator. In our architecture, the answer generator takes in the query and the retrieved documents to predict answers. We do not modify the structure of the answer generator, and thus the choice of answer generator is flexible. Our main contribution lies in the improvement of the retriever part - we improved the retrieval by proposing FLMR to replace DPR, and showed that better retrieval quality enhances the ultimate VQA performance. We demonstrate by experiments in Table 1 that both large language models (such as T5) and large multi-modal models (such as BLIP2) can benefit from FLMR. In this sense, if we choose PALI as the answer generator (if one day it is publicly available), we can also augment PALI with retrieved documents from the retriever. The PALI (retrieval-augmented)’s performance will very likely be improved when we upgrade DPR to FLMR.
>
>
> Thank you very much for your support for our work. We hope that our contribution is more clear to you after the rebuttal. Please don’t hesitate to ask follow-up questions if our explanation is not clear enough.

---

> > ### Comment · Reviewer_XziC · 2023-08-13
> > **Response**
> >
> > I would like to express my gratitude to the reviewers for their helpful responses. Based on that, I raise increased my score to 5.

---

### Official Review · Reviewer_ujV3 · 2023-07-26

**Soundness:** 3 good
**Presentation:** 4 excellent
**Contribution:** 3 good
**Rating:** 5
**Confidence:** 4

**Summary:**

This paper proposed an updated version of RA-VQA by incorporating a vision model aligned with text-based retriever in a late interaction way. Extensive experiments show the effectiveness of the proposed method. Thorough ablation studies and analysis were conducted.

**Strengths:**

1.	The paper is well-written and easy to read.
2.	Extensive experiments and analysis are provided to show the effectiveness of the proposed method.
3.	Experimental results demonstrate the superiority of the proposed method.


**Weaknesses:**

1.	Though the performance shows the superiority of the proposed method, the method of adding visual features is somewhat incremental compared to RA-VQA.
2.	Why is the performance of adding ROI&VE in w/o late-interaction setting (row 26 in Table 1) worse than w/o them (row 27)? This result is quite strange.
3.	It would be more intuitive to add the model size of each model in addition to the model names in Table 1.


**Questions:**

See the above weakness part.

---

> ### Author Rebuttal · Authors · 2023-08-08
>
> Thank you very much for your very positive review of our work. We are excited to learn that you enjoyed the presentation and the experiments.
>
> We would like to respond to your concerns as below:
>
> **1. Though the performance shows the superiority of the proposed method, the method of adding visual features is somewhat incremental compared to RA-VQA.**
>
> The original RA-VQA system did not consider dense image features in retrieval, and many recent KB-VQA systems (such as KAT) have similar limitations. We agree that it is an intuitive step to introduce dense image features, but how to incorporate these features in the context of multi-modal late interaction is non-trivial. We studied:
>
> - What image features are most beneficial for retrieval. We find that adding object-centric regional image representations is more effective compared to a single global representation and evenly split patches.
> - How to align the image and text modalities so that fine-grained cross-modality relevance can be assessed via late interaction. To this end, we propose a simple yet effective pre-training strategy that costs only 4 GPU hours to align the modalities under the late-interaction setting with a mapping network.
>
> Putting them together, we further show that FLMR can handle regional fine-grained image details with token-level interaction while DPR cannot, leading to superior retrieval and VQA performance. Please kindly refer to the general response for a more detailed listing of our technical contribution and novelty.
>
> **2. Why is the performance of adding ROI&VE in w/o late-interaction setting (row 26 in Table 1) worse than w/o them (row 27)? This result is quite strange.**
>
> - Thanks for your careful reading. This in fact reveals a desirable property of FLMR compared to DPR: incorporating ROI features improves the performance of FLMR, while this is not the case for DPR. We believe this is because the late-interaction mechanism allows FLMR to fully utilize the ROIs features, pin-pointing each of them to the relevant part of the document (Figure 3). In contrast, DPR’s one-dimensional embedding falls short of utilizing the rich ROI visual features.
>
>     These findings are supported by experimental results: The retriever at Row 26, Table 1 corresponds to DPR with visual encoder and region of interest features (Row 8, Table 2), while the one at Row 27 refers to the original DPR retriever at Row 5, Table 2. If we take a closer look at Table 2, we can observe that adding feature-based vision (Visual Encoder) improves performance (comparing Row 5 and Row 7). However, the performance is reduced when ROI features are incorporated (comparing Row 7 and Row 8). We discussed this behavior in Figure 2 and Lines 292-297. We will add one more line in the ablation study in Table 1 to demonstrate the point more clearly - DPR with VE without ROI: Recall@5=83.43, EM=60.18, VQA Score=59.21. Comparing this result to Row 27, Table 1 shows that introducing ROIs to DPR significantly harms performance. We will use the additional space to explain it more clearly if accepted.
>
>
> **3. It would be more intuitive to add the model size of each model in addition to the model names in Table 1.**
>
> Thank you for your kind suggestion. We had the model sizes in the table before but we removed them to fit the table into the page. We will try to reduce the font size in the camera-ready version if accepted.
>
> Finally, thank you for your valuable support for our work.

---

> > ### Comment · Reviewer_ujV3 · 2023-08-14
> > **Response**
> >
> > Thanks for the authors' rebuttal. I will keep my original score.

---

### Official Review · Reviewer_Bs2i · 2023-07-26

**Soundness:** 2 fair
**Presentation:** 3 good
**Contribution:** 2 fair
**Rating:** 5
**Confidence:** 3

**Summary:**

In this paper, the authors focus on RA-VQA framework to solve the KB-VQA task. Specifically, this paper proposed Fine-grained Late-interaction Multi-modal Retrieval (FLMR) to address the limitation of inaccurate image representations and one-dimensional embeddings. FLMR leveraged late interaction and multi-dimensional representations to capture fine-grained relevance to retrieval performance. The proposed methods achieve sota performance in the OK-VQA dataset compared with systems with similar parameter sizes.

**Strengths:**

The paper mainly focus on improving multimodal retrieval for the KR-VQA system, in detail, the proposed method uses an additional vision model to capture fine-grained features and align with text features instead of only using image-to-text features. The experiments show that the introduction of image representations from large vision models can lead to better retrieval and VQA performance which could benefit further research. The results on OK-VQA dataset and ablation study show the effectiveness of the proposed method.

**Weaknesses:**

This paper only reports results on a single dataset KR-VQA and thus didn't clearly show if the conclusion could be generalized to other related benchmarks.


**Questions:**

Have the authors performed experiments on other KB-VQA datasets or is that planned? If not, do authors have any particular reason for the missing of multiple datasets?

**Limitations:**

Yes.

---

> ### Author Rebuttal · Authors · 2023-08-08
>
> Thank you for your positive review of the strength of our work. We hope that you enjoyed reading our paper. Below we would like to respond to your major concern and question:
>
> 1. **Have the authors performed experiments on other KB-VQA datasets or is that planned? If not, do authors have any particular reason for the missing of multiple datasets?**
>
>     We have performed experiments on additional datasets, including those that are released very recently. We provide additional results on F-VQA [1] and preliminary results on Infoseek [2] in the general response to address your concern about generalizing to other datasets.
>
>     We chose OK-VQA for its relevance to our research focus on knowledge retrieval, as it is the largest and most challenging knowledge-based VQA dataset. The alternatives, such as F-VQA and A-OKVQA, are less suitable for evaluating our specific system due to their limitations. For instance, F-VQA's small size and biases and A-OKVQA's emphasis on visual reasoning rather than knowledge retrieval posed significant challenges. Please kindly refer to the general response for detailed rationales for our dataset choice.  We will also release the codebase which benchmarks our system on these datasets.
>
>
> We believe that the additional results and clarifications provided could address the concerns you raised, and we hope this strengthens the case for the acceptance of our paper.
>
> [1] FVQA: Fact-based Visual Question Answering
>
> [2] Can Pre-trained Vision and Language Models Answer Visual Information-Seeking Questions?

---

> > ### Comment · Reviewer_Bs2i · 2023-08-18
> >
> > Thanks for the additional experiments and response to my questions! I think the rebuttal basically address my concern so I decide to raise the score to borderline accept

---

### Official Review · Reviewer_d3hH · 2023-07-27

**Soundness:** 3 good
**Presentation:** 4 excellent
**Contribution:** 3 good
**Rating:** 5
**Confidence:** 3

**Summary:**

This paper focuses on retrieval augmented visual question answering and addresses two major limitations in previous RA-VQA. Specifically, they propose FLMR and overcome the limitations by obtaining image representations using a vision model. FLMR also encodes images and questions using multi-dimensional embeddings to capture finer-grained relevance between queries and documents. To verify the effectiveness, FLMR conducts extensive experiments on the OK-VQA dataset.

**Strengths:**

1. This paper solves the two limitations of previous RA-VQA. It is important for improving the effectiveness of RA-VQA.
2. This paper is well-organized and easy to follow.
3. The proposed model is effective with much less parameters than large models.

**Weaknesses:**

Actually, the novelty of this paper is somewhat trivial. The multi-modal late interaction has been proposed by ColBERT. Besides, using multi dimensional embeddings to improve retrieval is not very novel.

**Questions:**

1. The authors should further clarify the technical novelty of this paper.
2. In line 163-164, please explain the improvement and details of pre-training the mapping network.

---

> ### Author Rebuttal · Authors · 2023-08-08
>
> **Weakness and Question 1**:
>
> Please kindly refer to the general response where we provide a list of our technical novelties. If accepted, we will highlight these points more clearly in the camera-ready version following your suggestions.
>
> **Question 2: In line 163-164, please explain the improvement and details of pre-training the mapping network.**
>
> Thank you for your question and suggestion. We will try to accommodate all details in the main content using the additional page and set pointers at Line 163-164 to direct readers to the discussion of improvements (Line 284-287) and details of hyperparameters and architecture (currently in Appendix E, Line 40-45). We apologize for not being able to fit all practical details in the main content due to limited space.
>
> Here for your convenience, we paste the information that you might be interested in:
>
> - **Details**: For FLMR, we use $N_{vt} = 32$ visual tokens per image representation and $d_L = 128$. The mapping network consists of two fully-connected layers with tanh activation. The output of last layer is reshaped into $N_{vt} × d_L$ visual tokens. We use 1 Nvidia A100 (80G) for all experiments. The optimizer is Adam. We use a learning rate of 1e−4, a batch size of 30, and 2 gradient accumulation steps for around 8k steps.
> - **Improvement**: The improvement brought by pre-training is analyzed in Line 284-287: we observe that pre-training the mapping network for vision-language alignment is crucial for good performance. Without such pre-training, performance degrades to 85.71 Recall@5 (Table 2 Row 14) from 89.32 Recall@5 (Table 2 Row 13). We will set a hyperlink in Line 164 which directs readers to this analysis in revision. Thank you for pointing this out.

---

> > ### Comment · Reviewer_d3hH · 2023-08-21
> > **Response**
> >
> > Thanks for the response! My concern is basically addressed. I will keep my score.

---

### Author Rebuttal · Authors · 2023-08-08

We appreciate the time and efforts of all seven reviewers in assessing our paper. The positive feedback on our work is highly encouraging, and we have taken great care to respond to each reviewer individually. Here, we hope to provide direct and concise clarifications addressing the common concerns raised by more than one reviewer. In response to reviewers’ request, we provide new experimental results showing the generalizability of our technique. We also address the concerns of technical contribution and novelty.

1. Clarification on Technical contribution/Novelty (raised by Reviewer d3hH, Reviewer XziC)

    We highlight our technical novelties and summarize the reviewers’ consensus below:

    - **1) Late-interaction for knowledge retrieval with multi-modal query:** To our knowledge, our work is the first to extend multi-dimensional late interaction for knowledge-based visual question answering. We tackle complex multi-modal retrieval going beyond the previous focus on exploiting fine-grained token-level textual information [5] and single modal queries [6]. As noted by Reviewer d3hH, Bs2i, ujV3, XziC, shyj, jgsy,  xLzi, extensive experiments confirm the superiority of our approach in improving knowledge retriever for KB-VQA, surpassing previous baselines on the OK-VQA dataset (Line 244-254).
    - ****2) Alignment of textual and image modalities with cost-effective mapping network pre-training:**** Reviewer ujV3, XziC, and xLzi explicitly acknowledged the novelty of this approach. Our approach captures fine-grained, cross-modality relevance via token-level interactions. We perform simple yet effective alignment pre-training which costs only 4 GPU hours (Line 160-168). In addition, we highlight that the proposed system is flexible in that it does not limit the use of vision encoders and pre-trained retrievers. This allows the retrieval performance to grow with the development of general-purpose vision encoders and text retrievers.
    - **********3) Utilizing fine-grained image information with object-centric regional representation.********** Adding ROI-based image patches for finer-grained image understanding is conceptually straightforward. However, existing DPR-based retrieval systems cannot benefit from this additional information due to the limited expressiveness of one-dimensional embedding. In contrast, our experiment shows that FLMR is able to harness the regional image representation for better multi-modal retrieval thanks to the multi-dimensional embeddings and the late interaction mechanism of FLMR (Line 298-305) as noted by Reviewer d3hH, Bs2i, ujV3, shyj. This is an intuitive but non-trivial advance from the DPR baseline. The multi-modal late-interaction retrieval process can be visually interpreted (Figure 3, as noted by Reviewer xLzi).
    - **4) Public release of implementation:** We will release the codebase upon publication, which offers a complete solution for advancing multi-modal retrieval from single-vector text-only retrieval for KB-VQA. To the best of our knowledge, this will be the first publicly available implementation for multi-modal late-interaction KB-VQA retrievers.
2. Evaluation on other KB-VQA datasets (raised by Reviewer Bs2i and Reviewer jgsy)

    We agree that further evaluation on other KB-VQA datasets would be beneficial and have elaborated on our consideration in the main content (Line 188-190) and Appendix A: Limitations (Line 3-9). In sum:

    There are limited options in the current research community. Our primary focus was on knowledge retrieval and we evaluate the retrieval performance on OK-VQA, as it is considered the most challenging and largest KB-VQA dataset based on facts and concepts. Other KB-VQA datasets that are currently available are less suitable for the following reasons:

    (1) A-OKVQA is more focused on “visual reasoning” instead of “information seeking” (e.g. How many people will dine at this table? Answer: One). Only a minor portion (~18%) of its questions requires outside knowledge bases (reported by dataset authors in [2]). Therefore, it is not suitable for evaluating a knowledge retrieval system since most questions are not grounded in knowledge documents.

    (2) F-VQA is limited in size and quality. It only has 5,826 examples, with biases towards frequent answers (e.g. “person”) and questions, as demonstrated in [1]. Therefore, we refrain from using it for the main evaluation.

    Given these constraints, we evaluated FLMR on 2 additional knowledge bases: Wikipedia (Table 2) and WIT (Appendix H, Line 94-123).

    We further performed an analysis on F-VQA [3] on reviewers’ requests (Reviewer Bs2i and jgsy). The results are attached below and will be included in the appendices of the final version. All numbers are averaged across 5 splits and the training hyperparameters are kept the same as on OKVQA.

    Furthermore, we have gained early access and obtained promising preliminary results on a newly proposed dataset, Infoseek [4], from Google after the submission of this manuscript.

    These results further support the superiority of FLMR over DPR and establish the generalizability of FLMR.

    |  | FVQA [3] | Infoseek [4] |
    | --- | --- | --- |
    | Retriever | Recall@5 (Std.) | Recall@5 |
    | DPR (From RA-VQA) | 68.58 (0.01) | 45.33 |
    | FLMR (From RA-VQA-v2)  (Visual Encoder) (ours) | 70.88 (0.01) | 47.78 |
    | FLMR (From RA-VQA-v2)  (Visual Encoder + 10ROIs) (ours) | 72.37 (0.01) | 50.23 |

[1] FVQA 2.0: Introducing Adversarial Samples into Fact-based Visual Question Answering

[2] A-OKVQA: A Benchmark for Visual Question Answering using World Knowledge

[3] FVQA: Fact-based Visual Question Answering

[4] Can Pre-trained Vision and Language Models Answer Visual Information-Seeking Questions?

[5] ColBERTv2: Effective and Efficient Retrieval via Lightweight Late Interaction

[6] FILIP: Fine-grained Interactive Language-Image Pre-Training

---

### Decision · Program_Chairs · 2023-09-21

**Decision:**

Accept (poster)

**Comment:**

The submission focuses on the visual question answering task, where the authors proposed a retrieval-augmented visual question answering (RA-VQA) framework via Fine-grained Late-interaction Multi-modal Retrieval (FLMR). Compared to dense passage retrieval (DPR), the authors propose to capture fine-grained information with multi-dimensional embeddings, as inspired by prior work such as ColBERT. The proposed approach is evaluated on the OK-VQA benchmark and achieves competitive performance.

The reviewers found the submission to be well written, and the proposed solution to be well motivated. They found the empirical performance of the proposed method to be strong on the OK-VQA dataset, especially on knowledge retrieval performance, and that the authors provided extensive evaluations and analyses of the proposed method. However, some of the reviewers had concerns on (1) limited novelty, given the similarity of the late interaction to ColBERT, and (2) evaluations only on the OK-VQA dataset. The authors provided a rebuttal to address these concerns. Notably, they provided additional evaluations on FVQA and Infoseek benchmarks.

All reviewers responded to the authors' rebuttal, except for shyj, who is already in favor of accepting (weak accept) the paper. Among the six reviewers who acknowledged the rebuttal, five found the rebuttal to have addressed their concerns, and recommended borderline accepts (4 reviewers) and accept (1 reviewer). One reviewer decided to recommend borderline reject on the basis of the lacking of evaluation on A-OKVQA. The AC agrees with this reviewer that additional evaluations on A-OKVQA, even negative ones, would make the empirical evaluations more complete and informative, but also understands the rationales provided by the authors on why the adopted benchmarks are preferable than A-OKVQA. Overall, the AC believes the concerns on evaluation on more benchmarks to be adequately addressed.

The AC thus recommends acceptance of the submission, and encourages the authors to incorporate the additional experiments during the rebuttal, and the reviewers' feedback into the final version.